# Gradient Deconfliction via Orthogonal Projections onto Subspaces For Multi-task Learning

## Abstract

Although multi-task learning (MTL) has been a preferred approach and successfully applied in many real-world scenarios, MTL models are not guaranteed to outperform single-task models on all tasks mainly due to the negative effects of conflicting gradients among the tasks. In this paper, we fully examine the influence of conflicting gradients and further emphasize the importance and advantages of achieving non-conflicting gradients which allows simple but effective trade-off strategies among the tasks with stable performance. Based on our findings, we propose the Gradient Deconfliction via Orthogonal Projections onto Subspaces (GradOPS) spanned by other task-specific gradients. Our method not only solves all conflicts among the tasks, but can also effectively search for diverse solutions towards different trade-off preferences among the tasks. Theoretical analysis on convergence is provided, and performance of our algorithm is fully testified on multiple benchmarks in various domains. Results demonstrate that our method can effectively find multiple state-of-the-art solutions with different trade-off strategies among the tasks on multiple datasets.

## 1 Introduction

Multi-task Learning (MTL) aims at jointly training one model to master different tasks via shared representations and bottom structures to achieve better and more generalized results. Such positive knowledge transfer is the prominent advantage of MTL and is key to the successful applications of MTL in various domains, like computer visions (Misra et al., 2016; Kokkinos, 2017; Zamir et al., 2018; Liu et al., 2019), natural language processing (Dong et al., 2015; McCann et al., 2018; Wang et al., 2020; Radford et al., 2019), and recommender systems (Ma et al., 2018a;b; Tang et al., 2020; Wen et al., 2020). Many works also make further improvements via task-relationship modelling (Misra et al., 2016; Ma et al., 2018a; Xia et al., 2018; Liu et al., 2019; Tang et al., 2020; Xi et al., 2021) to fully exploit the benefit from shared structures. However, while such design allows positive transfer among the tasks, it also introduces the major challenges in MTL, of which the most dominating one is the conflicting gradients problem.

Gradients of two tasks, $g_i$ and $g_j$, are considered conflicting if their dot product $g_i \cdot g_j < 0$. With conflicting gradients, improvements on some tasks may be achieved at the expense of undermining other tasks. The most recent and representative works that seek to directly solve conflicts among the tasks are PCGrad (Yu et al., 2020) and GradVac (Wang et al., 2020). For each task $\mathcal{T}_i$, PCGrad iteratively projects its gradient $g_i$ onto gradient directions of other tasks and abandons the conflicting part; GradVac argues that each task pair $(\mathcal{T}_i, \mathcal{T}_j)$ should have unique gradient similarity, and chooses to modify $g_i$ towards such similarity goals. However, these methods only discussed convergence guarantee and gradient deconfliction under two-task settings (Liu et al., 2021a). For MTLs with 3 or more tasks, the properties of these algorithms may not hold because the sequential gradient modifications are not guaranteed to produce non-conflicting gradients among any two task pairs. Specifically, as demonstrated in Figure 1(b) and 1(c), the aggregated update direction $G'$ of these methods (e.g. PCGrad) still randomly conflicts with different original $g_i$ because of the randomly selected processing orders of task pairs, leading to decreasing performance on corresponding tasks.

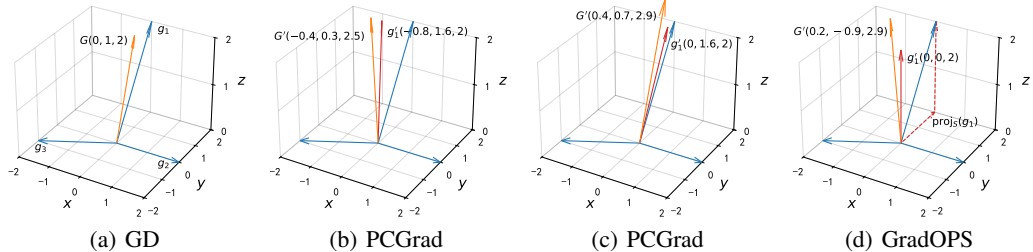

(a) GD  (b) PCGrad  (c) PCGrad  (d) GradOPS

Figure 1: Illustrative example of gradient conflicts in a three-task learning problem using gradient descent (GD), PCGrad and GradOPS. Task-specific gradients are labeled $g_1$, $g_2$ and $g_3$. The aggregated gradient $G$ or $G'$ in (a),(b) and (c) conflicts with the original gradient $g_3$, $g_2$ and $g_3$, respectively, resulting in decreasing performance of corresponding tasks. Note that different processing orders in PCGrad ([1,2,3] for (b), [3,2,1] for (c)) lead to conflicts of $G$ with different original $g_i$. In contrast, the GradOPS-modified $g_1'$ is orthogonal to $S = \text{span}\{g_2, g_3\}$ with the conflicting part on $S$ removed, similarly for $g_2'$ and $g_3'$ (omitted). Thus, neither each $g_i'$ nor $G'$ conflicts with any of $\{g_i\}$.

Another stream of work (Sener & Koltun, 2018; Lin et al., 2019b;a; Liu et al., 2021a) avoids directly dealing with gradient conflicts but casting the MTL problems as multi-objective optimization (MOO) problems. Though practically effective, applications of most MOO methods are greatly limited since these methods have to delicately design complex algorithms for certain trade-off among the tasks based on conflicting gradients. However, solutions toward certain trade-off may not always produce expected performance, especially when the convex loss assumption fails and thus convergence to Pareto optimal fails. In addition, different scenarios may require divergent preferences over the tasks, whereas most MOO methods can only search for one Pareto optimal toward certain trade-off because their algorithms are designed to be binded with certain trade-off strategy. Therefore, it's difficult to provide flexible and different trade-offs given conflicting gradients. Even for Liu et al. (2021a),Liu et al. (2021b) and Navon et al. (2022) which claim to seek Pareto points with balanced trade-offs among the tasks, their solutions might not necessarily satisfy the MTL practitioners' needs, since it's always better to provide the ability of reaching different Pareto optimals and leave the decision to users (Lin et al., 2019a).

In this paper, we propose a simple yet effective MTL algorithm: Gradient Deconfliction via Orthogonal Projections onto Subspaces spanned by other task-specific gradients (GradOPS). Compared with existing projection-based methods (Yu et al., 2020; Wang et al., 2020), our method not only completely solves all conflicts among the tasks with stable performance invariant to the random processing orders during gradient modifications, but is also guaranteed to converge to Pareto stationary points regardless of the number of tasks to be optimized. Moreover, with gradient conflicts among the tasks completely solved, our method can effectively search for diverse solutions toward different trade-off preferences simply via different non-negative linear combinations of the deconflicted gradients, which is controlled by a single hyperparameter (see Figure 2).

The main contributions of this paper can be summarized as follows:

- Focused on the conflicting gradients challenge, we propose a orthogonal projection based gradient modification method that not only completely solves all conflicts among the tasks with stable and invariant results regardless of the number of tasks and their processing orders, the aggregated final update direction is also non-conflicting with all tasks.

- With non-conflicting gradients obtained, a simple reweighting strategy is designed to offer the ability of searching Pareto stationary points toward different trade-offs. We also empirically testified that, with gradient-conflicts completely solved, such a simple strategy is already effective and flexible enough to achieve similar trade-offs with some MOO methods and even outperform them.

- Theoretical analysis on convergence is provided and comprehensive experiments are presented to demonstrate that our algorithm can effectively find multiple state-of-the-art solutions with different trade-offs among the tasks on MTL benchmarks in various domains.

Figure 2: Visualization of trade-offs in a 2D multi-task optimization problem. Shown are trajectories of each method with 3 different initial points (labeled with black ●) using Adam optimizer (Kingma & Ba, 2014). See Appendix B for details. GD is unable to traverse the deep valley on two of the initial points because there are conflicting gradients and the gradient magnitude of one task is much larger than the other. For MGDA (Sener & Koltun, 2018), CAGrad (Liu et al., 2021a), and IMTL-G (Liu et al., 2021b), the final convergence point is fixed for each initial point. In contrast, GradOPS could converge to multiple points in the Pareto set by setting different $\alpha$.

## 2 PRELIMINARIES

In this section, we first clarify the formal definition of MTL. Then we discuss the major problems in MTL and state the importance of achieving non-conflicting gradients among the tasks.

**Problem Definition.** For a multi-task learning (MTL) problem with $T > 1$ tasks $\{\mathcal{T}_1, ..., \mathcal{T}_T\}$, each task is associated with a loss function $\mathcal{L}_i(\theta)$ for a shared set of parameters $\theta$. Normally, a standard objective for MTL is to minimize the summed loss over all tasks: $\theta^* = \arg\min_\theta \sum_i \mathcal{L}_i(\theta)$.

**Pareto Stationary Points and Optimals.** A solution $\theta$ dominates another $\theta'$ if $\mathcal{L}_i(\theta) \leq \mathcal{L}_i(\theta')$ for all $\mathcal{T}_i$ and $\mathcal{L}_i(\theta) < \mathcal{L}_i(\theta')$ holds for at least one $\mathcal{T}_i$. A solution $\theta^*$ is called Pareto optimal if no solution dominates $\theta^*$. A solution $\theta$ is called Pareto stationary if there exists $\boldsymbol{w} \in \mathbb{R}^T$ such that $w_i \geq 0$, $\sum_{i=1}^T w_i = 1$ and $\sum_{i=1}^T w_i g_i(\theta) = \boldsymbol{0}$, where $g_i = \nabla_\theta \mathcal{L}_i(\theta)$ denotes the gradient of $\mathcal{T}_i$. All Pareto optimals are Pareto stationary, the reverse holds when $\mathcal{L}_i$ is convex for all $\mathcal{T}_i$ (Drummond & Iusem, 2004; Cruz et al., 2011). Note that MTL methods can only ensure reaching Pareto stationary points, convergence to Pareto optimals may not holds without the convex loss assumption.

**Conflicting Gradients.** Two gradients $g_1$ and $g_2$ are conflicting if the dot product $g_1 \cdot g_2 < 0$. Let $G$ be the aggregated update gradient, i.e. $G = \sum_{i=1}^T g_i$, we define two types of conflicting gradients:

- conflicts among the tasks if there exists any two task $\mathcal{T}_i$ and $\mathcal{T}_j$ with $g_i \cdot g_j < 0$,
- conflicts of the final update direction with the tasks if any task $\mathcal{T}_i$ satisfies that $G \cdot g_i < 0$.

For clarity, we introduce the concept of **strong non-conflicting** if $g_i \cdot g_j \geq 0, \forall i, j$ holds, and **weak non-conflicting** if the aggregated gradient $G$ dose not conflict with all original task gradients $g_i$. Note that strong non-conflicting gradients are always guaranteed to be weak non-conflicting, whereas weak non-conflicting gradients are not necessarily strong non-conflicting. Most existing MOO methods focus on seeking weak non-conflicting $G$ toward certain trade-offs, which is directly responsible for task performance. Current projection based methods (Yu et al., 2020; Wang et al., 2020) can only ensure strong non-conflicting gradients with $T = 2$, and when $T > 2$ even weak non-confliction is not guaranteed.

**Advantages of Strong Non-Conflicting Gradients.** Without strong non-conflicting gradients, MOO methods are limited to find only one solution toward certain trade-off by delicately balancing the conflicting gradients, which empirically may lead to unsatisfying results if such trade-off is less effective when convex assumption of $\mathcal{L}_i$ fails. In contrast, with strong non-conflicting gradients, we instantly acquire the advantage that it will be much easier to apply different trade-offs on $G$ simply via non-negative linear combinations of the deconflicted gradients, all guaranteed to be non-conflicting with each original $g_i$. With such ability, stable results with various trade-offs are also ensured since all tasks are always updated toward directions with non-decreasing performance.

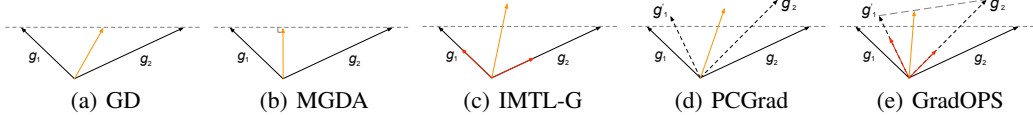


(a) GD     (b) MGDA     (c) IMTL-G     (d) PCGrad     (e) GradOPS


Figure 3: Visualization of the update direction (in yellow) obtained by various methods on a two-task learning problem. We rescaled the update vector to half for better visibility. $g_1$ and $g_2$ represent the two task-specific gradients. MGDA proposes to minimize the minimum possible convex combination of task gradients, and the update vector is perpendicular to the dashed line. IMTL-G proposes to make the projections of the update vector onto $\{g_1, g_2\}$ to be equal. PCGrad and GradOPS project each gradient onto the normal plane of the other to obtain $g_1'$ and $g_2'$. For PCGrad, the final update vector is the average of $\{g_1', g_2'\}$. GradOPS further reweights $\{g_1', g_2'\}$ to make trade-offs between two tasks. As a result, the final update direction of GradOPS is flexible between $g_1'$ and $g_2'$, covering the directions of MGDA and IMTL-G instead of been fixed as other methods, and always doesn't conflict with each task-specific gradient.

Thus, our GradOPS aims to solve MTL problems with the following goals: (1) to obtain stable and better performance on all the tasks by achieving strong non-conflicting gradients, (2) to provide simple yet effective strategies capable of performing different trade-offs.

## 3 METHOD

### 3.1 STRONG NON-CONFLICTING GRADIENTS

Though recent works (Yu et al., 2020; Wang et al., 2020) have already sought to directly deconflict each task specific gradient $g_i$ with all other tasks iteratively, none of theses methods manage to solve all conflicts among the tasks as stated in Section 1, let alone to further examine the benefits of strong non-conflicting gradients. As shown in Figure 1(b) and 1(c), both the modified $g_i'$ and the aggregated gradient $G'$ of existing methods may still conflict with some of the original task gradients $\{g_i\}$.

To address the limits of existing methods and to ensure strong non-conflicting gradients, GradOPS deconflicts gradients by projecting each $g_i$ onto the subspace orthogonal to the span of the other task gradients. Formally, GradOPS proceeds as follows: (i) For each $g_i$ in any permutation of the original task gradients, GradOPS first identifies whether $g_i$ conflicts with any of the other tasks by checking the signs of $g_i \cdot g_j$ for all $\{g_j\}_{j \neq i}$. (ii) If no conflict exists, $g_i$ remains unmodified: $g_i' = g_i$. Otherwise, GradOPS projects $g_i$ onto the subspace orthogonal to $S = \text{span}\{g_j\}_{j \neq i}$. The orthogonal basis for $S$ as $\{u_j\}$ is computed using the Gram-Schmidt procedure:

$$u_1 = g_1, \text{ if } i \neq 1 \text{ else } u_2 = g_2,$$
$$u_j = g_j - \sum_{k < j, k \neq i} \text{proj}_{u_k}(g_j), j > 1, j \neq i, \tag{1}$$

where $\text{proj}_u(v) = \frac{u \cdot v}{\|u\|^2} u$. Given the orthogonal basis $\{u_j\}$, the modified $g_i'$ is orthogonal to $S$:

$$g_i' = g_i - \sum_{j \neq i} \text{proj}_{u_j}(g_i). \tag{2}$$

Eventually, we have $g_i' \cdot g_j \geq 0, \forall i, j$ and thus each $g_i'$ does not conflict with every original $g_j$. (iii) With strong non-conflicting gradients $\{g_i'\}$, the vanilla aggregated update direction $G' = \sum_i g_i'$ is guaranteed to be non-conflicting with any of the original $g_j$ since $G' \cdot g_j = (\sum_i g_i') \cdot g_j = \sum_i (g_i' \cdot g_j) \geq 0, \forall j$, the same holds for any non-negative linear combinations of the $\{g_i'\}$.

With this procedure, GradOPS ensures that both each modified task gradient $g_i'$ and the final update gradient $G$ do not conflict with each original task gradient $g_j$, thus completely solves the conflicting gradients problem. In addition, unlike (Yu et al., 2020; Wang et al., 2020), our result is stable and invariant to the permutations of tasks during the procedure. Theoretical analysis on convergence of GradOPS is provided as **Theorem** 1 in Appendix A.1, under mild assumptions of neural network and small update step size.

### 3.2 TRADE-OFFS AMONG TASKS

---

**Algorithm 1** GradOPS Update Rule

---

**Require**: task number $T$, a constant $\alpha$, initial model parameters $\theta$

1: **while** not converged **do**
2:     $g_i \leftarrow \nabla_\theta \mathcal{L}_i(\theta), \forall i$
3:     **for** $i = 1; i \leq T$ **do**
4:         **if** $\exists k \neq i, g_i \cdot g_k < 0$ **then**
5:             compute orthogonal basis $\{u_j\}$ according to Eq. 1
6:             $g_i' = g_i - \sum_{j \neq i} \text{proj}_{u_j}(g_i)$
7:         **else**
8:             $g_i' = g_i$
9:         **end if**
10:     **end for**
11:     $G' = \sum_i g_i'$
12:     compute scaling factors $\{w_i\}$ according to Eq. 5
13:     update $\theta$ with gradient $G'_{\text{new}} = \sum_i w_i g_i'$
14: **end while**

---

As shown in Figure 2, most existing methods could only converge to one point in the Pareto set, without the ability of flexibly performing trade-offs among the tasks. Figure 3(a) to 3(d) give a more detailed illustration about the fixed update strategies of different methods, different solutions are achieved only through different initial points. However, note that MTL practitioners may need varying trade-offs among the tasks by demand in real-world applications (Lin et al., 2019a; Navon et al., 2021). In order to converge to points with different trade-offs, the most straightforward idea is to assign tasks different loss weights, i.e. to scale task gradients. However, this is often unsatisfying for two reasons: (1) update strategies of some methods like IMTL-G (Liu et al., 2021b) are invariant to gradients scales, and (2) it is difficult to assign appropriate weights without prior knowledge. Therefore, we further propose to dynamically scale the modified task gradients with strong non-conflicting guarantees obtained by GradOPS to adjust the update direction.

To this end, we first define $R_i$ as the scalar projection of $G' = \sum_i g_i'$ onto the original $g_i$:

$$R_i = ||G'|| \times \cos(\phi_{\langle g_i, G' \rangle}) = \frac{G' \cdot g_i}{||g_i||}. \tag{3}$$

Note that $R_i \geq 0$ as GradOPS always ensures $g_i \cdot G' \geq 0$. Given the same $G'$, each $R_i$ can be regarded as a measure of the angle between $g_i$ and $G'$, or a measure of the real update magnitude on direction of $g_i$, indicating the dominance and relative update speed of task $\mathcal{T}_i$ among the tasks. A new update direction $G'_{\text{new}} = \sum_i w_i g_i'$ is thus obtained by summing over the reweighted $g_i'$ with:

$$r_i = \frac{R_i}{\sum_i R_i / T}, \tag{4}$$

$$w_i = \frac{r_i^\alpha}{\sum_i r_i^\alpha / T}, \tag{5}$$

where $w_i$ functions as the scale factor calculated from $\{r_i\}$ and determines trade-offs among the tasks, $r_i$ reveals the relative update magnitude on direction $g_i$ and always splits the tasks into dominating ones $\{\mathcal{T}_i | r_i > 1\}$ and the dominated ones with $r_i \leq 1$. Thus, a single hyperparameter $\alpha \in \mathbb{R}$ on $r_i^\alpha$ will allow effective trade-offs among the dominating and dominated tasks via non-negative $w_i$ in Eq. 5. Details on convergence of GradOPS with different $w$ are analysed in **Theorem** 2.

**Discussion about $\alpha$.** With $\alpha > 0$, performance of the dominating tasks are expected to be improved since higher value of $\alpha$ will enforce $G'_{\text{new}}$ toward $\{g_i'\}$ of these tasks and yield greater $\{w_i\}$. While $\alpha < 0$, the dominated tasks are emphasized since lower value of $\alpha$ will ensure greater $\{w_i\}$ for tasks with smaller $\{r_i\}$. Specifically, a proper $\alpha < 0$ will pay more attention to the dominated tasks, and thus obtain more balanced update gradient, which is similar to IMTL-G. Keep decreasing $\alpha$ will further focus on tasks with smaller gradient magnitudes, which coincides with the idea

of MGDA. An example is provided in Figure 2, where the top right points in GradOPS($\alpha$=-2) and GradOPS($\alpha$=-5) converge to the similar points to IMTL-G and MGDA, respectively. Note that $G'_{\text{new}} = G'$ when $\alpha = 0$. A pictorial description of this idea is shown in Figure 3(e): Different $\alpha$ will redirect $G'_{\text{new}}$ between $g'_1$ and $g'_2$.

The complete GradOPS algorithm is summarized in Algorithm 1. Note that PCGrad is a special case of our method when $T = 2$ and $\alpha = 0$.

### 3.3 ADVANTAGES OF GRADOPS

Now we discuss the advantages of GradOPS. Both GradOPS and existing MOO methods can converge to Pareto stationary points, which are also Pareto optimals under convex loss assumptions. Thus ideally, there exists no dominating methods but only methods whose trade-off strategy best suited for certain MTL applications. Realizing that the assumptions may fail in practice and convergence to Pareto optimals may not hold, we are interested in achieving robust performance via simplest strategies instead of designing delicate trade-off strategies, which may fail to reach expected performance with violations of the assumptions.

Our Goal of securing robust performance is achieved via ensuring strong non-conflicting gradients. Though (Yu et al., 2020; Wang et al., 2020) are already aware of the importance of solving gradient conflicts (see Section 1), none of them manage to guarantee strong non-conflicting gradients, which is achieved by GradOPS. In addition, we further conclude empirically the benefit of achieving strong non-conflicting gradients, which is all fully exploited in GradOPS:

With all conflicts solved, effective trade-offs among the tasks are much easier. With conflicting gradients, existing methods have to conceive very delicate algorithms to achieve certain trade-off preferences. The trade-off strategy in Liu et al. (2021a) requires solving the dual problem of maximizing the minimum local improvements, IMTL-G (Liu et al., 2021b) applies complex matrix multiplications only to find $G$ with equal projection onto each $g_i$. Unlike these methods, GradOPS can provide solutions with different trade-offs simply via non-negative linear combinations of the deconflicted gradients, and achieves similar trade-offs with different MOO methods, like MGDA and IMTL-G, or even outperforms them. Moreover, GradOPS, which only requires tuning a single hyperparameter, also outperforms the very expensive and exhaustive grid search procedures. See Section 4 and Appendix C.3 for details on effects of tuning $\alpha$. Moreover, experiment results in Section 4 imply that a certain trade-off strategy may not always produce promising results on different datasets. Unlike most MOO methods which are binded with certain fixed trade-offs, GradOPS is also less affected and more stable since it flexibly supports different trade-offs and always ensures that all tasks are updated toward directions with non-decreasing performance.

Lastly, it's worth mentioning that GradNorm (Chen et al., 2018) can also dynamically adjust gradient norms. However, its target of balancing the training rates of different tasks can be biased since relationship of the gradient directions are ignored, which can be improved by taking real update magnitude on each task into account as we do in Eq. 3.

## 4 EXPERIMENTS

In this section, we evaluate our method on diverse multi-task learning datasets including two public benchmarks and one industrial large-scale recommendation dataset. For the two public benchmarks, instead of using the frameworks implemented by different MTL methods with various experimental details, we present fair and reproducible comparisons under the unified training framework (Lin & Zhang, 2022). GradOPS inherits the hyperparameters of the respective baseline method in all experiments, except for one additional hyperparameter $\alpha$. We include the additional experimental details and results in Appendix B and C respectively. Source codes will be publicly available.

**Compared methods.** (1) Uniform scaling: minimizing $\sum_i \mathcal{L}_i$; (2) Single-task: solving tasks independently; (3) existing loss reweighting methods (Chen et al., 2018), projection methods (Yu et al., 2020; Wang et al., 2020) and MMO methods (Sener & Koltun, 2018; Liu et al., 2021a;b).

**Evaluation.** In addition to common evaluation metrics in each experiment, we follow (Maninis et al., 2019; Liu et al., 2021a; Navon et al., 2022) and report two metrics to capture the overall

| Method | Income | Marital | Education | Average | $\Delta m\%$ | MR |
|---|---|---|---|---|---|---|
| Single-task | 0.9454 | 0.9817 | 0.8875 | 0.9382 | 0.00 | 4.33 |
| Uniform scaling | 0.9407 | 0.9763 | 0.8829 | 0.9333 | 0.52 | 13.00 |
| GradNorm | 0.9435 | 0.9829 | 0.8873 | 0.9379 | 0.03 | 6.00 |
| PCGrad | 0.9428 | 0.9812 | 0.8856 | 0.9365 | 0.18 | 11.00 |
| GradVac | 0.9436 | 0.9798 | 0.8856 | 0.9363 | 0.20 | 10.33 |
| MGDA | **0.9460** | 0.9833 | 0.8870 | 0.9388 | -0.06 | 3.00 |
| CAGrad | 0.9444 | 0.9820 | 0.8863 | 0.9376 | 0.07 | 7.00 |
| IMTL-G | 0.9443 | 0.9826 | 0.8864 | 0.9378 | 0.05 | 6.33 |
| GradOPS($\alpha$=0) | 0.9436 | 0.9817 | 0.8861 | 0.9371 | 0.12 | 9.33 |
| GradOPS($\alpha$=2) | 0.9432 | 0.9769 | 0.8864 | 0.9355 | 0.28 | 10.33 |
| GradOPS($\alpha$=-1) | 0.9443 | 0.9832 | 0.8866 | 0.9380 | 0.02 | 5.67 |
| GradOPS($\alpha$=-2) | 0.9453 | 0.9837 | 0.8871 | 0.9387 | -0.05 | 3.33 |
| GradOPS($\alpha$=-3) | 0.9459 | **0.9854** | **0.8876** | **0.9396** | **-0.15** | **1.33** |

Table 1: Experiment results on UCI Census-income dataset. The best scores are shown in bold and the second-best scores are underlined.

| Method | Segmentation | | Depth | | Surface Normal | | | | | $\Delta_m\% \downarrow$ | MR $\downarrow$ |
|---|---|---|---|---|---|---|---|---|---|---|---|
| | | | | | Angle Distance | | Within $t° \uparrow$ | | | | |
| | mIoU $\uparrow$ | Pix Acc $\uparrow$ | Abs Err $\downarrow$ | Rel Err $\downarrow$ | Mean $\downarrow$ | Median $\downarrow$ | 11.25 $\uparrow$ | 22.5 $\uparrow$ | 30 $\uparrow$ | | |
| Single-task | 28.46 | **55.78** | 0.6674 | 0.2839 | 30.39 | 23.68 | 24.56 | 47.98 | 59.76 | 0.00 | 4.67 |
| Uniform scaling | 27.50 | 53.31 | 0.6061 | 0.2627 | 32.80 | 27.61 | 19.85 | 41.53 | 53.68 | 6.50 | 11.33 |
| GradNorm | 26.54 | 52.69 | 0.5902 | 0.2500 | 31.40 | 26.00 | 21.62 | 44.02 | 56.29 | 3.10 | 7.78 |
| PCGrad | 27.53 | 53.51 | 0.5980 | 0.2606 | 32.61 | 27.32 | 20.10 | 42.00 | 54.11 | 5.72 | 9.67 |
| GradVac | 28.33 | 54.66 | 0.5908 | 0.2530 | 32.74 | 27.50 | 19.79 | 41.67 | 53.87 | 5.16 | 9.00 |
| MGDA | 18.57 | 46.99 | 0.7015 | 0.2913 | **29.83** | **23.27** | **24.84** | **48.63** | **60.58** | 5.64 | 6.78 |
| CAGrad | 27.07 | 54.14 | 0.5999 | 0.2598 | 30.93 | 25.35 | 22.30 | 45.09 | 57.34 | 1.93 | 7.67 |
| IMTL-G | 28.80 | 55.14 | 0.5954 | 0.2554 | 30.66 | 25.12 | 22.46 | 45.45 | 57.76 | 0.36 | 5.33 |
| GradOPS($\alpha$=0) | 27.08 | 54.40 | 0.5862 | 0.2545 | 32.12 | 26.86 | 20.69 | 42.69 | 54.89 | 4.32 | 8.11 |
| GradOPS($\alpha$=1) | 26.56 | 53.26 | 0.6055 | 0.2600 | 34.30 | 29.39 | 18.79 | 39.79 | 51.67 | 9.40 | 12.22 |
| GradOPS($\alpha$=-0.5) | **29.05** | 54.90 | **0.5819** | **0.2489** | 31.75 | 26.40 | 20.78 | 43.31 | 55.68 | 2.48 | 5.78 |
| GradOPS($\alpha$=-1) | 28.45 | 55.67 | 0.5895 | 0.2550 | 30.46 | 24.85 | 22.73 | 45.91 | 58.20 | **-0.23** | **4.44** |
| GradOPS($\alpha$=-1.5) | 27.34 | 54.71 | 0.6077 | 0.2509 | 30.34 | 24.80 | 22.75 | 45.94 | 58.34 | 0.43 | 5.11 |
| GradOPS($\alpha$=-3) | 20.75 | 48.28 | 0.6905 | 0.2782 | 29.87 | 23.75 | 24.10 | 47.79 | 59.94 | 4.73 | 7.11 |

Table 2: Experimental results on NYUv2 dataset. Arrows indicate the values are the higher the better ($\uparrow$) or the lower the better ($\downarrow$). Best performance for each task is bold, with second-best underlined.

performance: (1) Mean Rank (MR): The average rank of each method across the different tasks (lower is better). A method receives the best value, MR = 1, if it ranks first in all tasks. (2) $\Delta_m$: The average per-task performance drop of method $m$ with respect to the single-tasking baseline $b$: $\Delta_m = \frac{1}{T}\sum_{i=1}^{T}(-1)^{l_i}(M_{m,i} - M_{b,i})/M_{b,i}$ where $l_i = 1$ if a higher value is better for a criterion $M_i$ on task $i$ and 0 otherwise.

## 4.1 MULTI-TASK CLASSIFICATION

The UCI Census-income dataset (Dua & Graff, 2017) is a commonly used benchmark for multi-task learning, which contains 299,285 samples and 40 features extracted from the 1994 census database. Referring to the experimental settings in Ma et al. (2018a), we construct three multi-task learning problems from this dataset by setting some of the features as prediction targets. In detail, task Income aims to predict whether the income exceeds 50K, task Marital aims to predict whether this person's marital status is never married, and task Education aims to predict whether the education level is at least college. Since all tasks are binary classification problems, we use the Area Under Curve (AUC) scores as the evaluation metrics.

We summarize the experimental results for all methods in Table 1. The reported results are the average performance of 10 times experiments with random parameter initialization. As shown, GradOPS($\alpha$=0) outperforms the projection-based method PCGrad in all tasks, indicating the benefit of our proposed strategy which projects each task-specify gradient onto the subspace orthogonal to the span of the other gradients to deconflict gradients. Then, we compare the perfor-

| Method | Purchase | Cart | Wish | Average | $\Delta m\%$ | MR |
|--------|----------|------|------|---------|--------------|-----|
| Single-task | 0.8149 | 0.7616 | 0.8095 | 0.7953 | 0.00 | 7.33 |
| Uniform scaling | 0.8178 | 0.7606 | 0.8087 | 0.7957 | -0.04 | 9.00 |
| GradNorm | 0.8188 | 0.7609 | 0.8084 | 0.7960 | -0.08 | 6.67 |
| PCGrad | 0.8175 | 0.7609 | 0.8089 | 0.7958 | -0.05 | 8.67 |
| GradVac | 0.8182 | 0.7609 | 0.8087 | 0.7959 | -0.07 | 8.33 |
| IMTL-G | 0.8181 | 0.7602 | 0.8097 | 0.7960 | -0.08 | 8.00 |
| GradOPS($\alpha$=0) | 0.8190 | 0.7620 | 0.8095 | **0.7968** | **-0.19** | **3.33** |
| GradOPS($\alpha$=1) | 0.8186 | 0.7620 | 0.8067 | 0.7958 | -0.05 | 6.67 |
| GradOPS($\alpha$=2) | **0.8199** | **0.7625** | 0.8073 | 0.7966 | -0.15 | 4.33 |
| GradOPS($\alpha$=-0.5) | 0.8185 | 0.7618 | 0.8101 | **0.7968** | -0.18 | 4.00 |
| GradOPS($\alpha$=-1) | 0.8187 | 0.7616 | 0.8099 | 0.7967 | -0.17 | 4.33 |
| GradOPS($\alpha$=-2) | 0.8178 | 0.7604 | **0.8117** | 0.7966 | -0.16 | 7.33 |

Table 3: Experiment Results on the real large-scale recommendation system. The best and runner up results in each column are bold and underlined, respectively.

mance of GradOPS with different $\alpha$ and other MOO methods. We note that GradOPS($\alpha$=-1) recovers CAGrad and IMTL-G, and GradOPS($\alpha$=-2) approximates MGDA. This results show that GradOPS with a proper $\alpha < 0$ can roughly recover the final performance of CAGrad, IMTL-G and MGDA. GradOPS($\alpha$=-3) achieves the best performance in terms of Marital, Education and Average. Although GradOPS($\alpha$=2) doesn't achieve as good experimental results as GradOPS with $\alpha = \{-1, -2, -3\}$ do, it outperforms Uniform scaling baseline. And we find that for this dataset, almost any value of $-10 \leq \alpha \leq 5$ will improve Average performance over Uniform scaling baseline (see Appendix C.2 for details). It suggests that the way to deconflict gradients proposed by us is beneficial for MTL.

## 4.2 Scene Understanding

NYUv2 (Silberman et al., 2012) is an indoor scene dataset which contains 3 tasks: 13 class semantic segmentation, depth estimation, and surface normal prediction. We follow the experiment setup from (Liu et al., 2019; Yu et al., 2020), and fix SegNet (Badrinarayanan et al., 2017) as the network structure for all methods.

The results are presented in Table 2. Each experiment is repeated 3 times over different random parameter initialization and the mean performance is reported. Our methods GradOPS($\alpha$=-1.5) achieves comparable performance with IMTL-G. GradOPS($\alpha$=-0.5) and GradOPS($\alpha$=-1) achieve the best $\Delta_m$ and MR respectively. As surface normal estimation owns the smallest gradient magnitude, MGDA focuses on learning to predict surface normals and achieves poor performance on the other two tasks, which is similar to GradOPS with a lower $\alpha = -3$. It is worth noting that although MGDA has overall good results on UCI Census-income dataset, yet it has not as good $\Delta_m$ on this dataset, which suggesting that it is hard to generalize to different domains.

## 4.3 Large-scale Recommendation

In this subsection, we conduct offline experiments on a large-scale recommendation system in Taobao Inc. to evaluate the performance of proposed methods. We collect an industrial dataset through sampling user logs from the recommendation system during consecutive 7 days. There are 46 features, more than 300 million users, 10 millions items and about 1.7 billions samples in the dataset. We consider predicting three post-click actions: purchase, add to shopping cart (Cart) and add to wish list (Wish). We implement the network as a feedforward neural network with several fully-connected layers with ReLU activation, and adopt AUC scores as the evaluation metrics.

Results are shown in Table 3. As wish prediction is the dominated task in this dataset, GradOPS with a negative $\alpha = \{-0.5, -1, -2\}$ show better performance than GradOPS($\alpha$=0) in wish task, and GradOPS with a positive $\alpha = \{1, 2\}$ achieve better performance in the other two tasks. Our proposed GradOPS can successfully find a set of well-distributed solutions with different trade-offs,

and it outperforms the single-task baseline where each task is trained with a separate model. This experiment also shows that our method remains effective in real world large-scale applications.

## 5 RELATED WORK

Ever since the successful applications of multi-task learning in various domains, many methods have been released in pursuit of fully exploiting the benefits of MTL. Early branches of popular approaches heuristically explore shared experts and task-relationship modelling via gating networks (Misra et al., 2016; Ma et al., 2018a; Xia et al., 2018; Liu et al., 2019; Tang et al., 2020; Xi et al., 2021), loss reweighting methods to ensure equal gradient norm for each tasks (Chen et al., 2018) or to uplift losses toward harder tasks (Guo et al., 2018; Kendall et al., 2018). GradOPS is agnostic to model architectures and loss manipulations, thus can be combined with these methods by demands.

Recent trends in MTL focus on gradient manipulations toward certain objectives. Chen et al. (2020) simply drops out task gradients randomly according to conflicting ratios among tasks. Javaloy & Valera (2021) focuses on reducing negative transfer by scaling and rotating task gradients. Existing projection based methods explicitly study the conflicting gradients problem and relationships among task gradients. Some methods more precisely alter each task gradient toward non-negative inner product (Yu et al., 2020) or certain values of gradient similarity (Wang et al., 2020) with other tasks, iteratively. GradOPS makes further improvements over these methods, as stated in Section 3.1.

Originated from (Désidéri, 2012), some methods seek to find Pareto optimals of MTL as solutions of multi-objective optimization (MOO) problems. Sener & Koltun (2018) apply the Frank-Wolfe algorithm to solve the MOO problem defined in Désidéri (2012); Lin et al. (2019b) choose to project the closed-form solutions of a relaxed quadratic programming to the feasible space, hoping to achieve paretion-efficiency. More recent methods seek to find Pareto optimals with certain trade-off among the tasks instead of finding an arbitrary one. Liu et al. (2021a) seeks to maximize the worst local improvement of individual tasks to achieve Pareto points with minimum average loss. IMTL-G (Liu et al., 2021b) seeks solutions with balanced performance among all tasks by searching for an update vector that has equal projections on each task gradient. Note that some methods have already been proposed to provide Pareto solutions with different trade-offs. Lin et al. (2019a) explicitly splits the loss space into independent cones and applies constrained MMO methods to search single solution for each cone; Mahapatra & Rajan (2020) provides the ability of searching solutions toward certain desired directions determined by the input preference ray $r$; Navon et al. (2021) trains a hypernetwork, with preference vector $r$ as input, to directly predict weights of a certain MTL model with losses in desired ray $r$. Comparison of GradOPS and MOO methods are discussed in Section 3.3.

Finally, as discussed in Yu et al. (2020), we also state the difference of GradOPS with gradient projection methods (Lopez-Paz & Ranzato, 2017; Chaudhry et al., 2018; Farajtabar et al., 2020) applied in continual learning, which mainly solve the catastrophic forgetting problem in sequential lifelong learning. GradOPS is distinct from these methods in two aspects: (1) instead of concentrate only on conflicts of current task with historical ones, GradOPS focus on gradient deconfliction among all tasks within the same batch simultaneously to allow mutually positive transfer among tasks, (2) GradOPS aims to provide effective trade-offs among tasks with the non-conflicting gradients.

## 6 CONCLUSION

In this work, focusing on the conflicting gradients challenge in MTL, we introduce the idea of strong non-conflicting gradients, and further emphasize the advantages of acquiring such gradients: it will be much easier to apply varying trade-offs among the tasks simply via different non-negative linear combinations of the deconflicted gradients, which are all guaranteed to be non-conflicting with each original tasks. Thus, solutions with stable performance toward different trade-offs can also be achieved since all tasks are updated toward directions with non-decreasing performance in each training step for all trade-off preferences. To fully exploit such advantages, we propose a simple algorithm (GradOPS) that ensures strong non-conflicting gradients, based on which a simple reweighting strategy is also implemented to provides effective trade-offs among the tasks. Comprehensive experiments show that GradOPS achieves state-of-the-art results and is also flexible enough to achieve similar trade-offs with some of the existing MOO methods and even outperform them.

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

# A   DETAILED DERIVATION

## A.1   PROOF OF THEOREM 1

**Theorem 1.** *Assume individual loss functions $\mathcal{L}_1, \mathcal{L}_2, ..., \mathcal{L}_T$ are differentiable. Suppose the gradient of $\mathcal{L}$ is L-Lipschitz with $L > 0$. Then with the update step size $t < \frac{2}{TL}$, GradOPS in Section 3.1 will converge to a Pareto stationary point.*

*Proof.* As $\mathcal{L}$ is differential and L-smooth, we can obtain the following inequality:

$$\mathcal{L}(\theta') \leq \mathcal{L}(\theta) + \nabla\mathcal{L}(\theta)^T(\theta' - \theta) + \frac{1}{2}\nabla^2\mathcal{L}(\theta)\|\theta' - \theta\|^2$$

$$\leq \mathcal{L}(\theta) + \nabla\mathcal{L}(\theta)^T(\theta' - \theta) + \frac{1}{2}L\|\theta' - \theta\|^2 \tag{6}$$

Plugging in the GradOPS update by letting $\theta' = \theta - tG'$, we can conclude the following:

$$\mathcal{L}(\theta') \leq \mathcal{L}(\theta) - \left(tG \cdot G' - \frac{1}{2}Lt^2\|G'\|^2\right)$$

(Expanding, using the identity $G = \sum_i g_i, G' = \sum_i g'_i$)

$$= \mathcal{L}(\theta) - t\left(\sum_i g_i \cdot \sum_i g'_i - \frac{1}{2}Lt\|\sum_i g'_i\|^2\right)$$

$$= \mathcal{L}(\theta) - t\left(\sum_{i,j}(g_i \cdot g'_j) - \frac{1}{2}Lt\sum_{i,j}(g'_i \cdot g'_j)\right)$$

(Using the inequality $g_i \cdot g'_j \geq 0, \forall i, j$ from Section 3.1)

$$\leq \mathcal{L}(\theta) - t\left(\sum_i(g_i \cdot g'_i) - \frac{1}{2}Lt\sum_{i,j}(g'_i \cdot g'_j)\right)$$

(Using the inequality $g'_i \cdot g'_j \leq \|g'_i\| \cdot \|g'_j\| \leq \frac{1}{2}(\|g'_i\|^2 + \|g'_j\|^2)$)

$$\leq \mathcal{L}(\theta) - t\left(\sum_i(g_i \cdot g'_i) - \frac{1}{2}Lt\sum_{i,j}\frac{1}{2}(\|g'_i\|^2 + \|g'_j\|^2)\right)$$

$$= \mathcal{L}(\theta) - t\left(\sum_i(g_i \cdot g'_i) - \frac{1}{2}TLt\sum_i\|g'_i\|^2\right)$$

(Using the identity $g_i \cdot g'_i = \|g'_i\|^2, \forall i$)

$$= \mathcal{L}(\theta) - t\left(\sum_i\|g'_i\|^2 - \frac{1}{2}TLt\sum_i\|g'_i\|^2\right)$$

$$= \mathcal{L}(\theta) - t(1 - \frac{1}{2}TLt)\sum_i\|g'_i\|^2. \tag{7}$$

Note that $t(1-\frac{1}{2}Ltn)\sum_i\|g'_i\|^2 \geq 0$ when $t \leq \frac{2}{TL}$. Further, when $t < \frac{2}{TL}$, $t(1-\frac{1}{2}TLt)\sum_i\|g'_i\|^2 = 0$ if and only if $\|g'_i\|^2 = 0, \forall i$, i.e. $g'_i = \mathbf{0}, \forall i$.

Hence repeatedly applying GradOPS process with $t < \frac{2}{TL}$ can reach some point $\theta^*$ in the optimization landscape where $g'_i = \mathbf{0}, \forall i$. According to Section 3.1, we have $g_i = \mathbf{0}$ or $g_i$ belongs to subspace $S = \text{span}\{g_j\}_{j\neq i}$. This means that there exists a convex combination of the gradients $\{g_i\}$ at this point $\theta^*$ that equals zero, and therefore $\theta^*$ is Pareto stationary under the definitions in Section 2.

$\square$

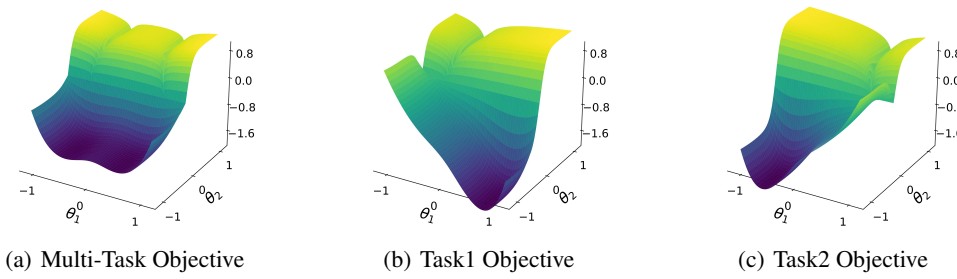

(a) Multi-Task Objective     (b) Task1 Objective     (c) Task2 Objective

Figure 4: Visualization of the loss surfaces of Figure 2.

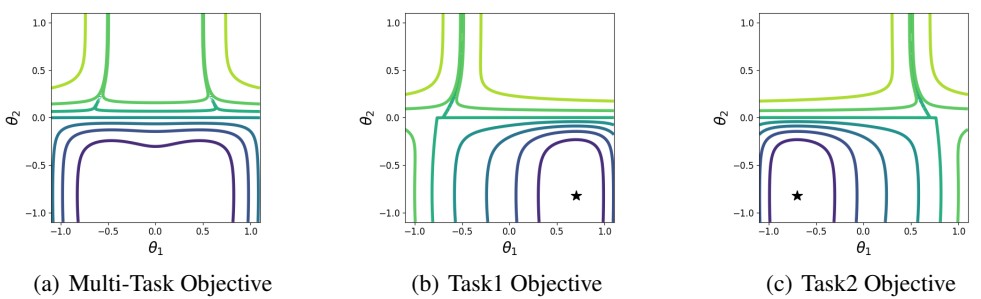

(a) Multi-Task Objective     (b) Task1 Objective     (c) Task2 Objective

Figure 5: Two-dimensional contour graphs of the three-dimensional surfaces in Figure 4. ★ denotes the point with lowest single-task loss.

## A.2    PROOF OF THEOREM 2

**Theorem 2.** *Assume individual loss functions $\mathcal{L}_1, \mathcal{L}_2, ..., \mathcal{L}_T$ are differentiable. Suppose the gradient of $\mathcal{L}$ is L-Lipschitz with $L > 0$. Then with the update step size $t < \min_{i,j} \frac{2w_i \|g_i'\|^2}{TL w_j^2 \|g_j'\|^2}$, GradOPS in Section 3.2 will converge to a Pareto stationary point.*

*Proof.* Similar to Section A.1, plugging in Inequality 6 by letting $\theta' = \theta - tG'_{\text{new}}$, we can conclude the following:

$$\mathcal{L}(\theta') \leq \mathcal{L}(\theta) - \left( tG \cdot G'_{\text{new}} - \frac{1}{2} L t^2 \|G'_{\text{new}}\|^2 \right)$$

$$\text{(Expanding, using the identity } G = \sum_i g_i, G'_{\text{new}} = \sum_i w_i g_i' )$$

$$= \mathcal{L}(\theta) - t \left( \sum_i g_i \cdot \sum_i w_i g_i' - \frac{1}{2} L t \| \sum_i w_i g_i' \|^2 \right)$$

$$\leq \mathcal{L}(\theta) - t \left( \sum_i w_i \|g_i'\|^2 - \frac{1}{2} T L t \sum_i w_i^2 \|g_i'\|^2 \right). \tag{8}$$

If $g_i' = \mathbf{0}, \forall i$, GradOPS reaches the Pareto stationary point. Otherwise, we denotes $\mathcal{T}^+$ as the tasks set where $\|g_i'\|^2 > 0, i \in \mathcal{T}^+$. Therefore, $w_i > 0, \forall i \in \mathcal{T}^+$ according to Section 3.2. In this case, with $t < \min_{i \in \mathcal{T}^+, j} \frac{2w_i \|g_i'\|^2}{TL w_j^2 \|g_j'\|^2}$, GradOPS can reach some point $\theta^*$ where $g_i' = \mathbf{0}, \forall i$, i.e. the Pareto stationary point. $\qquad\square$

## B  IMPLEMENTATION DETAILS

**Illustrative Example.**   We provide here the details for the illustrative example of Figure 2. We modify the illustrative example in (Liu et al., 2021a) and consider $\theta = (\theta_1 + \theta_2) \in \mathbb{R}^2$ with the following individual loss functions:

$$
\begin{aligned}
&\mathcal{L}_1(\theta) = c_1 f_1(\theta) + c_2 f_1(\theta) \text{ and } \mathcal{L}_2(\theta) = c_1 f_2(\theta) + c_2 f_2(\theta), \text{ where} \\
&f_1(\theta) = \log(\max(|5(-\theta_1 - 0.7) - \tanh(-3 * \theta_2)|, 0.0005)) + 1 \\
&f_2(\theta) = \log(\max(|5(-\theta_1 + 0.7) - \tanh(-3 * \theta_2)|, 0.0005)) + 1 \\
&g_1(\theta) = (1.5 * \tanh(2 * (-\theta_1 + 0.7)^2) * (\theta_1^2 + 1) + (-\theta_2 - 0.8)^2) - 2.5 \\
&g_2(\theta) = (1.5 * \tanh(2 * (-\theta_1 - 0.7)^2) * (\theta_1^2 + 1) + (-\theta_2 - 0.8)^2) - 2.5 \\
&c_1(\theta) = \max(\tanh(5 * \theta_2), 0) \text{ and } c_2(\theta) = \max(\tanh(-5 * \theta_2), 0).
\end{aligned}
$$

The multi-task objective is $\mathcal{L}(\theta) = \mathcal{L}_1(\theta) + \mathcal{L}_2(\theta)$. The three-dimensional loss surfaces and the corresponding two-dimensional contour graphs are shown in Figure 4 and 5, respectively. We pick 3 initial parameter vectors $\theta \in \{(-0.85, 0.75), (-0.85, -0.3), (0.9, 0.9)\}$ and performed 20,000 gradient updates to minimize $\mathcal{L}$ using the Adam optimizer with learning rate 0.001. The corresponding optimization trajectories with different methods is shown in Figure 2.

**Multi-Task Classification.**   In the UCI Census-income dataset, there are 199,523 training examples and 99,762 test examples. Following Ma et al. (2018a), we further randomly split test examples into a validation dataset and a test dataset by the fraction of 1:1. We apply a 2-layer fully-connected ReLU-activated neural network with 192 hidden units of each layer for all methods. We also place one dropout layer (Hinton et al., 2012) with $p = 0.3$ for regularization. For GradNorm (Chen et al., 2018) and CAGrad (Liu et al., 2021a) baselines which have additional hyperparameters, we follow the original papers and search $\alpha \in \{0.5, 1.5, 2.0\}$ for GradNorm and $c \in \{0.1, 0.2, ..., 0.9\}$ for CAGrad with the best average performance of the three tasks on validation dataset ($\alpha = 1.5$ for GradNorm and $c = 0.5$ for CAGrad). We train each method for 200 epochs with batch-size of 1024, and the Adam optimizer with a learning rate of $1e - 4$.

**Scene Understanding.**   We follow the training and evaluation procedure used in previous work on MTL (Liu et al., 2019; Yu et al., 2020) and apply SegNet (Badrinarayanan et al., 2017) as the network structure for all methods. For GradNorm (Chen et al., 2018) and CAGrad (Liu et al., 2021a) baselines, we follow the hyperparameters settings in the original papers with $\alpha = 1.5$ for GradNorm and $c = 0.4$ for CAGrad. Each method is trained for 200 epochs with a batch size of 2, using the Adam optimizer with a learning rate of $1e-4$. The learning-rate is halved to $5e-5$ after 100 epochs.

**Large-scale Recommendation.**   We collect an industrial dataset through sampling user logs from the recommendation system in Taobao during consecutive 7 days. There are 46 features, more than 300 million users, 10 millions items and about 1.7 billions samples in the dataset. We implement the network as a feedforward neural network with 4 fully-connected layers with ReLU activation. The hidden units are fixed for all models with hidden size $\{1024,512,256,128\}$. All methods are trained with Adam optimizer with a learning rate of $1e - 3$.

## C  ADDITIONAL EXPERIMENTS

### C.1  EFFECTS OF TUNING $\alpha$.

To better understand how $\alpha$ affects the $\{w_i\}$, we show the traces of $w_i$ during training for different values of $\alpha$ on UCI Census-income dataset in Figure 6. The positive and negative $\alpha$ have opposite effects on $w_i$ of tasks Income and Marital compared to the zero $\alpha$. And the $\alpha$ with larger absolute values further widen the gap.

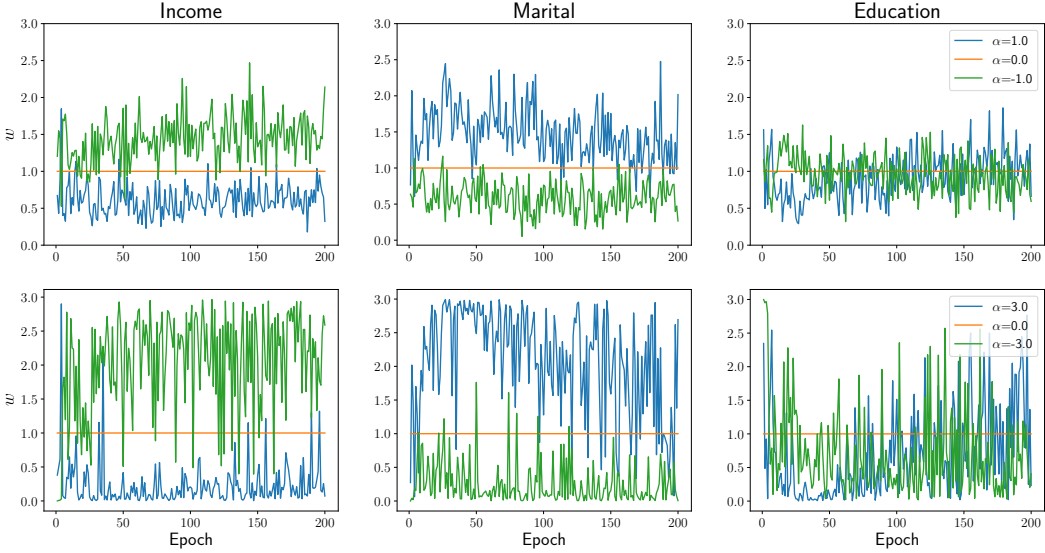

Figure 6: Traces of how $w_i$ change during training for different values of $\alpha$ on UCI Census-income dataset. On the top row, we show the comparison between $\alpha \in \{1, 0, -1\}$ and on the bottom row, we show the comparison between $\alpha \in \{3, 0, -3\}$. For $\alpha = 0$, $w_i$ remains at 1 for all three tasks. The positive $\alpha = 1.0$ assigns the dominated task Income a greater $w_i > 1$, and dominating task Marital a lower $w_i < 1$. And a larger value of $\alpha$ pushes weights farther apart. While, a negative $\alpha$ has the opposite effect.

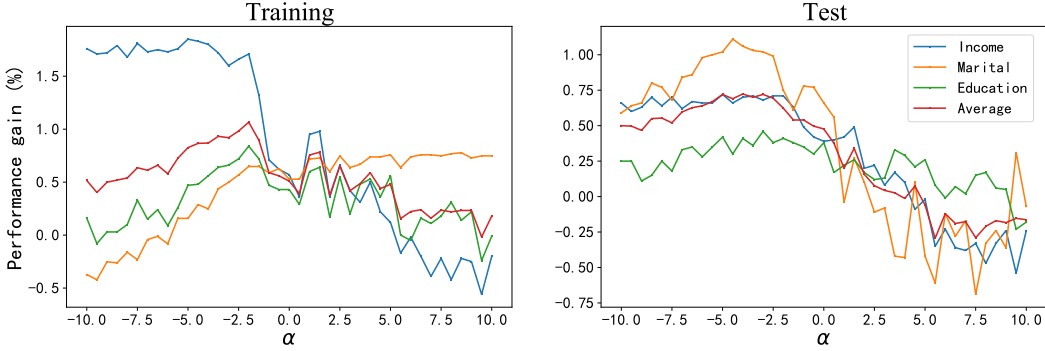

Figure 7: Performance gains with different $\alpha$ on UCI Census-income dataset. We show both the training and test performance gains compared to Uniform scaling baseline across all three tasks and the Average performance metrics. We enumerate $\alpha \in [-10, 10]$ in steps of 0.5.

## C.2  TRAINING AND TEST PERFORMANCE GAINS WITH DIFFERENT $\alpha$.

To test whether the performance of GradOPS are robust against the hyperparameter $\alpha$ changes, we show both the training and test performance gains with different $\alpha$ on UCI Census-income dataset in Figure 7. We note that the training performance gain of dominating task Marital rises as the value of $\alpha$ gets larger, while the gain of the dominated task Income goes down. This again proves the ability of GradOPS to obtain solutions with different trade-offs. It should be mentioned that higher training performance does not necessarily guarantee higher test performance. And the test performance of task Marital with $\alpha > 0$ is worse than the performance with $\alpha \leq 0$, which is inconsistent with the observation of corresponding training performance. The reason may be that the model overfits on task Marital, which suggesting a better regularization scheme for this domain. Note that we achieve Average performance gains on both training and test for almost all values of

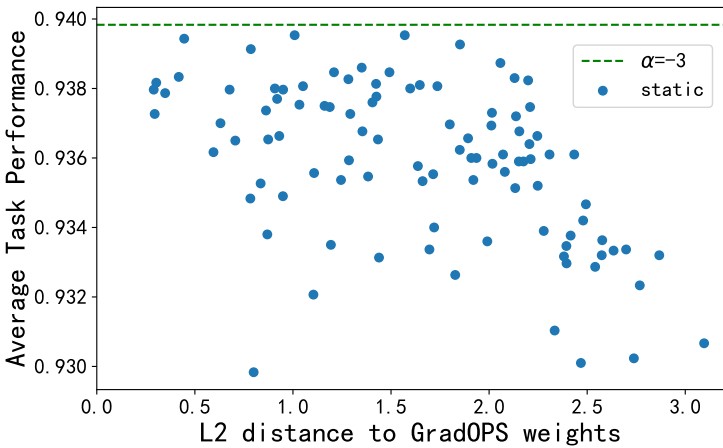

Figure 8: Performance comparison between GradOPS($\alpha$=-3) and GradOPS-static with grid search weights $w_i^{\text{static}}$. The x-axis denotes the L2 distance between $w_i^{\text{static}}$ and the average weights of GradOPS($\alpha$=-3) over training steps.

$-10 \leq \alpha \leq 5$, which indicating that GradOPS is numerically stable. Moreover, the consistently positive performance gains across all these values of $\alpha$ suggest that the way to deconflict gradients introduced by GradOPS can improve MTL performance.

### C.3   COMPARISON BETWEEN GRADOPS AND GRADOPS WITH GRID-SEARCH WEIGHTS.

To further verify the effectiveness of the strategy introduced in Section 3.2 for obtaining $w_i$, we conduct a comparative experiment where we train GradOPS with static $w_i^{\text{static}}$ (referred to as GradOPS-static) on UCI Census-income dataset. $\{w_i^{\text{static}}\}$ are considered as hyperparameters, and are sampled from candidates generated by grid search. The sum of $\{w_i^{\text{static}}\}$ is guaranteed to equal $T = 3$, and $G'_{\text{static}} = \sum_i w_i^{\text{static}} g'_i$ is used as the final update gradient. Then, we compare the performance of GradOPS-static to our GradOPS($\alpha$=-3).

The results are shown in Figure 8. We train GradOPS-static 100 times with different combinations of $\{w_i^{\text{static}}\}$. Even after this exhaustive grid search, GradOPS-static still falls short of our GradOPS($\alpha$=-3). Therefore, GradOPS can find the optimal grid search weights in one single training run.

