# OpenReview forum: "Gradient Deconfliction via Orthogonal Projections onto Subspaces For Multi-task Learning"
_ICLR.cc/2023/Conference — Submitted to ICLR 2023_

### Official Review · Reviewer_tv3z · 2022-10-23

**Confidence:** 4
**Correctness:** 3
**Technical Novelty And Significance:** 3
**Empirical Novelty And Significance:** 3
**Recommendation:** 6

**Clarity, Quality, Novelty And Reproducibility:**

Clarity: not bad.
Quality: the writing can be improved.
Novelty and Reproducibility: good.


**Details Of Ethics Concerns:**

I have no ethics concerns.

**Strength And Weaknesses:**

Strengths:
1. The proposed algorithm is simple and effective. I am very happy to see that such simple algorithm can achieve non-conflicting gradients and improved performance for multi-task learning problems.
2. Figure 1 is a good example to illustrate the differences between the proposed GradOPS and existing MTL algorithms.
3. Theoretical analysis of the convergence of GradOPS is provided, ensuring that the output of GradOPS can converge to a Pareto stationary point.
4. Authors also give a full discussion of the advantages of the proposed GradOPS in Section 3.3.
5. I find almost no typos in the paper.

Weaknesses and Suggestions:
1. More clarity on Figure 1. I appreciate the illustration of Figure 1. However, I believe that it can be improved in terms of the clarity and beauty. For instance, the caption of Figure 1 in Page 2 should include the calculation of the aggregated gradient G = \sum_{i=1}^{3}g_{i}, instead of letting readers seek the detailed aggregation rule of G in Subsection 3.1 in Page 4. Besides, different arrows with different colors in Figure1 represents different gradients, and it will be better to clarify them in the caption. Finally, the notation of GradOPS-modified g_{1}^{‘} looks not good enough. It should be g_{1}^{‘} instead of g^{‘}_{1} in Figure 1(b)(c)(d), to keep notation consistence with g_{1} in Figure 1(a).
2. The theoretical results should be placed in the main paper, and in particular, the assumptions of the convergence of the proposed algorithm in Theorem 1 & 2 should be expressed explicitly in the main paper (e.g. in the bottom of page 4). For example, the convergence of GradOPS in Theorem 1 requires the Lipschitz properties of the gradient and the small step size of GD. Such assumptions are quite essential to derive the final theoretical results, and should be clarified in the main paper (e.g. at least with the claim “the convergence of GradOPS is provided in Theorem 1, under mild assumptions of neural network and the step size of GD”). A counterexample is that, if we use ReLU activation function in deep neural network, the smooth assumption of neural network could not be satisfied and the convergence of GradOPS in Theorem 1 could not be guaranteed.
3. As far as I can tell, the proof of Theorem 1 and Theorem 2 is easy (and hence is correct), so the derived results should not be stated as Theorem. It will be better to call them as Proposition.
4. More explanations on the architecture of the deep neural networks (DNN) in experiment Part. As far as I can see, this paper uses different deep neural networks as backbones for different datasets, and authors also point out the name of these networks in the main paper. However, the components of DNN is very important in our experiments, and so I will suggest the authors to give more explanations on the components of DNN in the Appendix C (e.g. the kernel size of convolutional network, and the activation function in DNN).
5. Other minor suggestions: (1) Eqs.4 and 5 can be merged and put in one line for concise. (2) Avoid orphan line before Section 3.1. in Page 4 and the orphan line at the top of Page 9. (3) Typos: in caption of Figure 3, “the final updated direction”.


**Summary Of The Paper:**

This paper proposes a simple and effective gradient deconfliction algorithm, called GradOPS, for multi-task learning (MTL). Concretely, GradOPS projects the gradient associated with one task onto the subspace orthogonal to the span of the other task-specific gradients, achieving non-conflicting gradients for different tasks. Theoretical analysis of the convergence of GradOPS is provided, followed by extensive experiments on several multi-task learning datasets. Overall, this paper demonstrates the effectiveness of the proposed novel algorithms, both from theoretically and practically.

**Summary Of The Review:**

Thank authors for their detailed responses. In terms of others’ review and authors’ feedback, I remain my score as “weak accept.”

---

> ### Author Response · Authors · 2022-11-18
> **Author Response**
>
> Thank you very much for your comprehensive review and valuable feedback. We address your comments one by one as following:
>
>  > **Q1: More clarity and beauty on Figure 1.**
>
> Thank you for your careful and detailed advice on improving clarity and beauty of our paper and especially suggestions on illustration and presentation of Figure 1, which we really put a lot effort to make our demonstration as clear as possible. We completely agree with all your advice and we will revise our paper accordingly. Notations of g_{1}^{'} are already updated in the latest version. Other improvements like inclusion of explaining $G = \sum_{i=1}^{3}g_{i}$, due to limit in time and maximum allowed page numbers, will be done in future versions.
>
> > **Q2: Advice on theoretical analysis.**
>
> Your advice and rigorousness in theoretical analysis are really appreciated.
>
> 1\) The theoretical results should be placed in the main paper.
>
> Currently, that our theorems are placed in Appendix is our compromise to meet the requirement on page nums. We will try to place our theorems results in main paper in the future revision.
>
> 2\) Explicitly statements on assumptions of convergence of our theorem.
>
> Already done in latest updates.
>
> > **Q3: Theorem or Proposition.**
>
> Although the proof of our theorems is easy and simple, we would like to keep as them as Theorem because of the importance and irreplaceable roles of these two theorems in our paper.
>
> > **Q4: More explanations on the architecture of the deep neural networks (DNN) in experiment Part.**
>
> Details on architectures of DNN in our experiments are included in Appendix B. In latest version of our paper, readers are notified of such information at the beginning of Section 4 Experiments.
>
> > **Q5: Other minor suggestions.**
>
> 1\) Merging Eqs.4 and 5.
>
> Note that Eq.4 is needed to introduce $r_i$, which is used to identify the dominating and dominated tasks and is essential in later discussions. We find it necessary to keep both Eqs.4 and 5 for clarity.
>
> 2\) Avoid orphan lines.
>
> Thanks for your noticing, we will try our best to avoid such lines in the final version.
>
> 3\) "the final updated direction" vs "the final update direction"
>
> Figure 3 demonstrates the difference among the directions to be updated by each methods, thus we choose to use "update direction" instead of "updated direction".

---

> > ### Comment · Reviewer_tv3z · 2022-12-12
> > **Feedback**
> >
> > Thanks for the rebuttal. My concerns have been addressed to somewhat extent. I would keep my rating.

---

### Official Review · Reviewer_CAmc · 2022-10-24

**Confidence:** 4
**Correctness:** 1
**Technical Novelty And Significance:** 3
**Empirical Novelty And Significance:** 2
**Recommendation:** 3

**Clarity, Quality, Novelty And Reproducibility:**

**Clarity** I find the paper well written in general. However, writing could be improved a bit, e.g., by replacing expressions such as "What's more."

Moreover, I find quite concerning the usage of the word "gradient" along the entire text, as it is not clear to me that this modified gradients are still gradients. I also find some statements too strong for my taste (without further justification or citation), such as "Existing MOO methods only seek weak non-conflicting G towards certain trade-offs, _which is directly responsible for task performance._"

**Quality** Disregarding all the concerns I've presented in the section above, I find the quality of the paper ok.

**Novelty** The proposed method is rather novel.

**Reproducibility** The experiments are not reproducible since there is no code provided and the proposed method is not well-defined.

**Strength And Weaknesses:**

**Strengths**
- S1: I like the distinction between strong and weak non-conflicting gradients, although I find the explanations a bit too hand wavy for my taste.
- S2: The proposed method does not depend on the task ordering, which I find quite important.
- S3: Using scalar projections to measure the "dominance" of one task in the training process is quite interesting.
- S4: I really like the use of Mean Rank to compare methods in the experiments.

**Weaknesses**

Proposed method
- W1: I have serious concerns understanding the proposed method and the intuition behind it:
  - As of Eq. 1, $u_i$ is not defined for $i \neq 1$.
  - For $i \neq 1$, $u_2 = 0$ always.
  - It is not clear to me why this method should work. For example, if there are more tasks than parameters, then the basis generated by GS would span the entire parameter space. Then Eq. (2) would produce $g_i = 0$ for all tasks, unless for some subtle reason in the algorithm, $u_i$ is always different from 0.
   - I fail to understand the reason behind the double normalization in Eq. 4 and 5, rather than weighting $R_i$ directly in Eq. 4.
- W2: One of the advantages of GradOPS is that "it solves all conflicts among tasks," meaning that the resulting vectors $g'_i$ hold $\langle g'_i, g_j \rangle \geq 0$ for all $i$ and $j$. I fail to see this as a milestone by itself, since this could very well be achieved by multiplying all gradients by 0. Why is this important? And why are the new update directions meaningful?
- W3: While the introduction of the parameter $\alpha$ is introduced with the advantage of exploring more solutions, there is no control or intuition for the practitioner on which values of $\alpha$ set to get to a desired trade-off.

Experiments
- W4: While the experiments are repeated ten times, there are no standard deviations and no best averaged results were marked in bold, rather than running any statistical test to check for significant changes.
- W5: Results on the NYUv2 dataset are significantly worse than those reported in the literature, for example, in RotoGrad [1] and RLW [2].

Literature review:
- L1: Literature review is missing some conflicting gradient methods that do not fit in the MOO vs. projection discussion: e.g. GradDrop [3] randomly drops elements of the task gradients, and RotoGrad [1] applies rotation matrices in the heads to align task gradients.
- L2: Some comments feel a bit off. For example, saying that IMTL-G is a state-of-the-art method is rather questionable.

**Questions:**
- Q1. How were the MR and $\Delta m$ computed? Individually for each run, or over the averaged results?
- Q2: Which values of $\alpha$ were selected for CAGrad and GradNorm?

[1] - Spotlight, ICLR 2022 - [RotoGrad: Gradient Homogenization in Multitask Learning](http://arxiv.org/abs/2103.02631)

[2] - ArXiv - [A Closer Look at Loss Weighting in Multi-Task Learning](http://arxiv.org/abs/2111.10603)

[3] -  NeurIPS 2021 - [Just Pick a Sign: Optimizing Deep Multitask Models with Gradient Sign Dropout](https://proceedings.neurips.cc//paper_files/paper/2020/hash/16002f7a455a94aa4e91cc34ebdb9f2d-Abstract.html)

**Summary Of The Paper:**

Training multitask learning models can be difficult at times due to the existence of conflicting gradients. That is, when computing the gradient of the model's parameters, different tasks present different gradients which can differ in magnitude and direction, counteracting each other when added up, and leading to parameter updates that obtain unsatisfactory results.

In this paper, the authors argue the necessity and importance of the absence of gradient conflict during training, and provide a method to explicity ensure this setting. The method, GradOPS, generates a new set of gradients by first transforming the original gradients into an orthogonal set of vectors, and then subtracting their contribution (i.e. projection) from the original gradients, making sure that the resulting gradients do not conflict the original ones. Finally, these gradients are added up, weighted by their contribution to the final update direction (and a hyperparameter).

The authors then conduct three experiments on a multitask binary-classification, NYUv2, and a recommendation system, proving their method effective compared with other MTL approaches.


**Summary Of The Review:**

Overall, I find the idea of the paper (robustifying, in some sense, projection-based methods to improve conflicting gradients) appealing, and it could be a nice direction to pursue. The paper also have some cool reflections regarding conflicting gradients.

However, the proposed method has some questionable design choices and technical flaws that need further investigation/clarification, including the motivation of why the proposed procedure leads to sensible directions to follow during optimization.

This, added to some concerns on the experimental part, makes me believe that the manuscript needs a bit more of work, and hence leans me towards rejection.

---

> ### Author Response · Authors · 2022-11-18
> **Author Response Part 3, responses to L1, L2 and Q1, Q2**
>
> > **L1: Comparison with non-projection based gradient deconflicting methods.**
>
> Our paper is updated and these methods (GradDrop and RotoGrad) are referenced and discussed in Paragraph 2 of Section 5 Related Work. Here we would like to provide more details and discussions.
>
> 1\) We believe GradOPS is more generally applicable compared with GradDrop and RotoGrad for two reasons:
>
> - With strong non-conflicting gradient ensured, GradOPS is capable of searching different Pareto stationary points toward different trade-offs via a simple yet flexible re-weighting strategy. However, for RotoGrad, neither convergence to Pareto stationary points is guaranteed, nor the ability of allowing flexible trade-offs among the tasks is supported.
>
> - RotoGrad mainly focuses on situations where models have a shared backbone which produces a common latent representation across tasks. This indicates that they are mostly effective when the inputs for different tasks are shared and only the prediction output differs. In contrast, projection based methods like GradOPS and PCGrad, and MOO methods like MGDA and CAGrad are more generally applicable, reguardless of the model architecture or task inputs as long as there are shared structures.
>
> 2\) Empirically, experiment results of GradDrop and RotoGrad (Rotate Only) on UCI dataset are provided, we can see that GradOPS still outperforms these methods.
>
> |                        | Income | Marital | Education | Average |
> |------------------------|--------|---------|-----------|---------|
> | Single-task            | 0.9454 | 0.9817  | 0.8875    | 0.9382  |
> | GradDrop               | 0.9431 | 0.9805  | 0.8855    | 0.9364  |
> | RotoGrad (Rotate Only) | 0.9430 | 0.9816  | 0.8854    | 0.9367  |
> | GradOPS (alpha=0)      | 0.9436 | 0.9817  | 0.8861    | 0.9371  |
> | GradOPS (alpha=-3)     | 0.9459 | 0.9854  | 0.8876    | 0.9396  |
> ||
>
>
> > **L2: Some comments feel a bit off. For example, saying that IMTL-G is a state-of-the-art method is rather questionable.**
>
> We agree with your opinion. To be more rigorous, we already revised our statements on IMTL-G and avoided describing IMTL-G as SOTA in our paper. Comparisons with more recent methods like RotoGrad is also provided in our response to L1, and GradOPS still outperforms these methods.
>
> > **Q1. How were the MR and Δm computed? Individually for each run, or over the averaged results?**
>
> MR and Δm are computed over the averaged results, details on calculation are demonstrated in the last paragraph before Section 4.1.
>
> > **Q2: Which values of α were selected for CAGrad and GradNorm?**
>
> Such experimental details about how we determine values of hyperparameters for GradNorm and CAGrad are updated and included in Appendix B as follows:
>
> - On UCI dataset, we follow the original papers and search $\alpha \in \\{0.5, 1.5, 2.0\\}$ for GradNorm, and $c \in \\{0.1, 0.2, ..., 0.9\\}$ for CAGrad. Hyperparameters with best average performance of the three tasks on validation dataset are selected ($\alpha=1.5$ for GradNorm and $c=0.5$ for CAGrad).
>
> - On NYUv2 dataset, we follow the hyperparameters settings in the original papers with $\alpha=1.5$ for GradNorm and $c=0.4$ for CAGrad.
>
> Readers are now notified of such information in Appendix B at the beginning of Section 4 Experiments.
>
> > **Other Concerns on "Clarity, Quality, Novelty And Reproducibility":**
>
> - Thanks for the suggestion, we will avoid frequently using expressions like "What's more", and modify some of the statements that are too strong for more rigorousness.
>
> - Regarding your concern about reproducibility, a primal version of GradOPS source code is provided  in the Supplementary Material. Formal version of both GradOPS and experiments will be released later.
>
> &nbsp;
>
> Please let us know if you have any comments, questions, or concerns in light of this clarification. Thank you!

---

> > ### Author Response · Authors · 2022-12-09
> > **We are expecting your valuable feedback**
> >
> > Dear Reviewer CAmc,
> >
> > We believe we’ve addressed all your concerns including:
> > - detailed explaination and demonstration of our algorithm
> > - detailed information on hyperparameters of baseline methods and experiment metrics as well as tests of statistical significance on experiments results
> > -  additional experiments with recent methods
> >
> > We’d be grateful if you could share your further feedback. Thank you for your valuable time and consideration.
> >
> > Regards, Authors

---

> > > ### Comment · Reviewer_CAmc · 2022-12-11
> > > **Reply post-rebuttal**
> > >
> > > Dear authors,
> > > I am sorry for the delayed reply, let me reflect my thoughts after your last reply.
> > >
> > > I still have concerns regarding the soundness of the approach. I will try to keep it short, and go over each of the points on your reply for ease of exposition:
> > >
> > > - **W1.1)** The updated equation (1) *is still wrong*. Now for any $j != 1$ we have $u_2 = g_2$ and $u_1$ is not defined (if you drop $j>1$ then $u_1 = g_1$, which is also strange). This does not correspond with your demonstration in that same reply.
> > > - **W1.2)** First off, if you were aware of this limitation, it should've been mentioned in the manuscript. Second, my concern is not about how rare this case is, but rather on the problems on the methodology that raises. This degenerated case depends **only** on the number of tasks and parameters, and not in the gradients themselves. This means that, even in the case where you have almost perfectly aligned task gradients, if there are too many of them, all the gradients will become zero with GradOPS, which does not seem like something that should happen.
> > > - **W1.3)** I'll gloss over this point. I still think you could've integrated the $\alpha$ in Eq. 4 directly.
> > > - **W2** I can buy this argument, yet I still feel they these claims are not properly supported.
> > > - **W3** My concern was mainly about the lack of semantics on tuning $\alpha$, but I can live with the way it is right now. Something I still don't understand is what you mean with "by checking $r_i > 1$, as $r_i$ changes on each iteration step and depends on all the prior training trajectory.
> > > - **W4, W5, L1**. My concerns with statistical significance, error bars, etc., regard all the experiments, not only UCI. I don't understand why you only updated that table (but not with the results from answer L1), and the results from answer W5 are not included in the updated version neither. I also don't understand why comparing with Rotate Only which does not scale any gradient and is clearly a lesser version of RotoGrad.
> > >
> > > I therefore still have some major concerns regarding: i) the soundness of the proposed approach, as it is not clear which directions it follows/what is the proposed method semantically doing; as well as ii) the effectiveness of the algorithm and the significance of the results, as it seems that more resources were put into the proposed algorithm than on the others methods, and that tuning $\alpha$ provides random trajectories to follow (there is no correlation in the results as you change $\alpha$) and thus a bigger chance to obtain a good result.
> > >
> > > I am not changing my score. I hope the extra feedback is helpful to the authors for the next revision of the work.

---

> > > > ### Author Response · Authors · 2022-12-12
> > > > **Short Response to Reply post-rebuttal**
> > > >
> > > > Thank you for your feedback on our rebuttal, here are our short replies on major concerns.
> > > >
> > > > > **W1.1**: It seems you still have some misunderstandings on our algorithms and especially Eq(1).
> > > > More specifically, step 1 in Eq(1) means that: when $i=1$(we are determining $g'_1$), $u_1$ would not be needed and $g_1'$ can be determined by $g_1$ and other $u_j(j\neq 1)$, and thus $u_2$ can be the first to be generated, without loss of generality; when $i\neq 1$, it's ok to generate $u_1$ first and $u_i$ is not needed to calculate $g'_i$. In the first version, $u_i$ refers to the $i-th$ generated $u$ and is not necessarily bond with $g_i$, which may leads to mis-understandings. To avoid such mis-understandings, related descriptions in our paper are already updated to a more rigorous version during the first rebuttal stage. We believe the current version and our example in Author Response Part 1 is straight and clear enough to demonstrate the entire flow of GradOPS.
> > > >
> > > > > **W1.2**:  Discussions on how existing methods including GradOPS behave under such extreme cases will be added in later versions. Note that some existing methods shared the same problems and no discussions are provided for reference.
> > > >
> > > > > **W4, W5**: As we already stated in our rebuttal, the inclusion of these information dose not affect our existing results and conclusions in our paper. Thus such detailed results with statistical significance are provided as supplemental materials to make our statements and results in paper more clear, readers who are interested in such details are free to check our supplemental materials.
> > > >
> > > > > **L1**: RotoGrad consists of two parts: Scale Only and Rotate Only, which are independent and can be applied separately, similar to IMTL-L and IMTL-G. Since Scale Only RotoGrad and IMTL-L focus on loss weights and can be combined with many methods including GradOPS by demand, we choose to compare with Rotate Only RotoGrad and IMTL-G instead of introducing functions like Scale Only RotoGrad/IMTL-L to GradOPS and other methods for fair comparison. Moreover, previous papers include Nash-MTL and RotoGrad also only compare with IMTL-G instead of IMTL.
> > > >
> > > > > **Other concerns on the effectiveness of the algorithm and the significance of the results, as it seems that more resources were put into the proposed algorithm than on the others methods**:
> > > > We also put comparable resource looking for hyperparameters of compared methods to ensure best performance of these methods, as stated in Experiments. We believe it's not necessary to report every search results of other methods since these are unrelated with our topic. However, details on performance of GradOPS with different $\alpha$ are provided because it's necessary to demonstrate the effectiveness of making trade-offs among the tasks via tuning $\alpha$
> > > >
> > > >
> > > > Thank you again for your reply, even if you choose to keep your score, we still hope that our response will address some of your concerns.

---

> ### Author Response · Authors · 2022-11-18
> **Author Response Part 2, responses to W3 - W5**
>
> > **W3: discussions on tuning $\alpha$ in practice.**
>
> Discussions on how $\alpha$ can make different trade-offs among the tasks are presented at the end of Section 3.2 and testified empirically in Appendix C. The major points are:
>
> - By checking if $r_i > 1$ or not, all tasks can be classified as dominating or dominated ones. Generally speaking, applying positive $\alpha$ will enforce the dominating tasks, and negative $\alpha$ will produce more balanced performance and enhance the dominated tasks (see Figure 6 in Appendix C).
>
> - In experiments, performance of using different $\alpha$ are presented to mainly demonstrate empirically the effectiveness of trade-offs among the tasks of GradOPS. Results and conclusions on how $\alpha$ yields different trade-offs are all consistent with our analysis in Section 3.2 as well as among the experiments in Section 4.
>
> - We would also like to emphasize that such simple and effective trade-offs among the tasks via non-negative linear combinations of task gradients are only available given strong non-conflicting gradients. As commented by other reviewers: "It is the flexible re-weighting that makes it interestingly competitive".
>
> - We also believe that ensuring strong non-conflicting gradients and studying more delicate and efficient trade-off strategies shall be an interesting and promising future research direction in MTL.
>
> > **W4: statistical significance of experiment results.**
>
> Results of experiments on UCI dataset are appended with standard deviations as well as significance of improvements under hypothesis testings. Methods that achieve significant improvements over Single-task results are highlighted with grey background (a paired one-sided t-test with confidence threshold of 0.05). Best averaged results are already marked in bold. Note that the inclusion of these information dose not change our statements and conclusions in Experiments.
>
> > **W5: performance on the NYUv2 dataset differs from those reported in other papers.**
>
> Results among different papers vary mostly because they apply different settings. Since PCGrad serves as an importance baseline method, we keep the same settings with PCGrad without data augmentation techniques, which is the major reason of the significant difference in results on the NYUv2 dataset with other papers. In addition, we also repeated the same experiments with data augmentation and noticed significant performance improvements for all methods, and the experimental phenomena and conclusions are similar and consistent with existing results in our paper.

---

> ### Author Response · Authors · 2022-11-18
> **Author Response Part 1, responses to W1 and W2**
>
> We thank the reviewer for the valuable feedback. We address the reviewer's concerns in the following.
>
> > **W1: Your concerns can be summarized into the following three questions. Here are our clarifications in details, respectively:**
>
> **1\) Clarification and detailed explainations of GradOPS.**
>
> Thanks you for pointing out that our description on the GradOPS deconflicting procedure can be ambiguous. Related descriptions in our paper are already updated to a more rigorous version. For simplicity and without loss of generality, we will bind $u_j$ with $g_j$ in a more rigorous version of Eq. (1) and (2):
>
> - Eq. (1):
>
>     $u_1 = g_1, \text{ if $i \neq 1$ else } u_2 = g_2$
>
>     $u_j = g_j - \sum_{k<j, k \neq i}{{\rm proj}_{u_k}(g_j)}, j > 1, j \neq i$
>
> - Eq. (2):
>
>     $g_i' = g_i - \sum_{j \neq i}{{\rm proj}_{u_j}(g_i)}$.
>
> &nbsp;
>
> Here is also an example for detailed demonstration of how $\\{g_i'\\}$ are generated given $T=3$:
>
> step1\) calculate $g_1'$, $i=1$:
>
> - if $g_1 \cdot g_2 \geq 0$ and $g_1 \cdot g_3 \geq 0$ then $g_1' = g_1$
> - else:
>
>     - According to the first line in Eq.(1): start with $j=2$, $u_2=g_2$
>     - According to the second line in Eq.(1): $u_3 = g_3 - proj_{u_2}(g_3)$
>     - According to Eq.(2): $g_1' = g_1 - (proj_{u_2}(g_1) + proj_{u_3}(g_1))$
>
>
>
> step2\) calculate $g_2'$, $i=2$:
>
> - if $g_2 \cdot g_1 \geq 0$ and $g_2 \cdot g_3 \geq 0$ then $g_2' = g_2$
> - else:
>     - According to the first line in Eq.(1): start with $j=1$, $u_1=g_1$
>     - According to the second line in Eq.(1): $u_3 = g_3 - proj_{u_1}(g_3)$
>     - According to Eq.(2):  $g_2' = g_2 - (proj_{u_1}(g_2) + proj_{u_3}(g_2))$
>
> step3\) calculate $g_3'$, $i=3$:
>
> - if $g_3 \cdot g_1 \geq 0$ and $g_3 \cdot g_2 \geq 0$ then $g_3' = g_3$
> - else:
>     - According to the first line in Eq.(1): start with $j=1$, $u_1=g_1$
>     - According to the second line in Eq.(1): $u_2 = g_2 - proj_{u_1}(g_2)$
>     - According to Eq.(2):  $g_3' = g_3 - (proj_{u_1}(g_3) + proj_{u_2}(g_3))$
>
> **2\) Effectiveness of GradOPS when the number of tasks is greater than the number of parameters.**
>
> Your concern about the effectiveness of GradOPS when given more tasks than parameters is rather reasonable, since in such cases $g_i'=0$ for all $i$ will be much more likely to happen. We are already aware of such potential ineffectiveness of GradOPS, and for now choose to leave it as our future works mainly because such extreme case rarely happens in real applications. In common MTL cases, the number of parameters are expected to be much greater than task numbers, e.g. models with 1,000+ parameters for no more than 10 tasks. In such common cases, non-trivial $\\{g_i'\\}$ is expected to be pervasive. We also confirmed that on all three experiments in GradOPS, $g_i'=0$ is not observed.
>
> **3\) Purpose of the double normalization.**
>
> The normalization in Eq.(4) is important since $r_i$ is necessary to identify the dominating and dominated tasks by checking if the corresponding $r_i > 1$ or not. The normalization in Eq.(5) is designed to enable different scaling for flexible trade-offs among the dominating and dominated tasks by introduce $\alpha$. Please refer to our response to W3  for details on how $r_i$ and $\alpha$ allows such effective trade-offs.
>
> > **W2: importance of achieving non-trivial $\\{g_i'\\}$ and meaningfulness of the new update directions.**
>
> 1\) Firstly, GradOPS seeks to generate non-trivial and strong non-conflicting $\\{g_i'\\}$ that satisfies $g_i' \cdot g_j \geq 0, \forall i,j$ with at least one $g_i' \neq \\textbf{0}$. Such $\\{g_i'\\}$ can be achieved following our GradOPS procedure.
>
> 2\) Secondly, acquisition of non-trivial and strong non-conflicting $\\{g_i'\\}$ is the key to enable further effective trade-offs among the tasks, since every non-negative linear combinations (trade-offs) of the deconflicted gradients is guaranteed to be non-conflicting with each original $g_i$. Thus, all tasks are always updated toward directions with non-decreasing performance for all trade-offs.

---

### Official Review · Reviewer_FaLj · 2022-10-26

**Confidence:** 3
**Correctness:** 3
**Technical Novelty And Significance:** 3
**Empirical Novelty And Significance:** 3
**Recommendation:** 5

**Clarity, Quality, Novelty And Reproducibility:**

This paper is well-organized and somewhat novel; however, there are some weaknesses required to address.

**Strength And Weaknesses:**

Strengths:

1. The underlying idea of this work is well-motivated. It hypothesizes a reasonable theory about the “conflicting gradients” issue in multi-task learning and follow-up with a simple algorithm.
2. The concept definition of “strong non-conflicting” and “weak non-conflicting” provide a clear description of the core issue, which in turn strongly serves as the basis for method design.
3. The proposed method is simple and general, it can be easily applied to various gradient-based approaches. Furthermore, the paper writing is professional and rigorous.

Weaknesses:

1. Although this work provides a novel projection-based approach, the core idea does not improve too prominently from previous work (Yu et al., 2020; Wang et al., 2020), in fact, PCGrad can also be generalized to the case of T > 2 as well (and the experimental results did not significantly exceed PCGrad, shown in Table 1, 2). Moreover, compared to PCGrad, the theoretical analysis of this paper is relatively weak.
2. The experimental results are somewhat disappointing, which makes the proposed method less convincing. For instance, the results in Table 1 and Table 2 are not exactly better than the baseline method (MGDA). In addition, the performance of GradOPS shown in Table 3 is highly dependent on a good choice of α values, which may mean that the method is not so general for different types of datasets.
3. There is no comparison with the state-of-the-art algorithm, in fact, the NashMTL (Navon et al., 2022) has surpassed the previous method, suggesting that the authors supplement the corresponding comparison experiments.

**Summary Of The Paper:**

This manuscript proposes a novel projection-based method for tackling the issues of gradient conflict in the multi-task learning problem, GradOPS. The work improves previous projection-based methods like PCGrad by projecting conflict gradients onto an orthogonal subspace. In this way, it can not only resolve all conflicts in more than two tasks but also effectively searches for diverse solutions with different trade-off preferences among different tasks.

**Summary Of The Review:**

This paper is well-organized and somewhat novel; however, there are some weaknesses required to address.

---

> ### Author Response · Authors · 2022-11-18
> **Author Response Part 2, responses to Q2 and Q3**
>
> > **Q2: Significance of GradOPS vs MGDA in Table1&2 and Discussion on GradOPS performance in Table3.**
>
> 1\) **GradOPS vs MGDA in Table 1 and Table 2**
>
> Note that MGDA improves performance of weak task at the expense of other tasks performance, the overall performance of MGDA on all task is relatively less competing, which is also discussed in CAGrad [2]. Specifically, GradOPS far outperforms MGDA on other tasks (e.g. AUC gain +0.0021 on Marital in Table 1) while still achieves comparable performance on weak tasks (AUC gain -0.0001 on Income in Table 1), the overall achievement of GradOPS on △m and MR in Table 2 also significantly outperfroms MGDA.
>
> 2\) **GradOPS performance in Table3**
>
> Here performance of GradOPS with different $\alpha$ are presented to mainly demonstrate the effectiveness of performing different trade-offs using our $\alpha$ strategy, given strong non-conflicting gradients. It's reasonable and expected that different datasets should apply different $\alpha$ for best performance since the underlying relationships and trade-offs among the tasks may vary. How to choose $\alpha$ in application is also fully discussed and empirically demonstrated in Section 3.2 and Appendix C.
>
> The major point is that GradOPS provides simple, straightforward and effective strategy to allow flexible trade-offs, which is achieved by obtaining strong non-conflicting gradients. As commented by other reviewers, "It is the flexible re-weighting that makes it interestingly competitive".
>
> Finally, we would also like to mention that GradOPS with different $\alpha$ already achieves top2 ranking on both the Average AUC and all task AUCs.
>
> > **Q3: Comparison with SOTA algorithm like NashMTL.**
>
> Results of NashMTL on UCI dataset is added and presented below. The trade-off strategy behind NashMTL is similar to that of IMTL-G and NashMTL can be regarded as a better version of IMTL-G. GradOPS still outperforms NashMTL on Income, Education and Average AUC.
>
> |                    | Income | Marital | Education | Average |
> |--------------------|--------|---------|-----------|---------|
> | IMTL-G             | 0.9443 | 0.9826  | 0.8864    | 0.9378  |
> | NashMTL            | 0.9440 | 0.9863  | 0.8865    | 0.9389  |
> | GradOPS (alpha=-3) | 0.9459 | 0.9854  | 0.8876    | 0.9396  |
> ||
>
> &nbsp;
>
> [1] Modeling task relationships in multi-task learning with multi-gate mixture-of-experts, KDD 2018
>
> [2] Conflict-Averse Gradient Descent for Multi-task Learning, NIPS 2021

---

> ### Author Response · Authors · 2022-11-18
> **Author Response Part 1, responses to Q1**
>
> Thank you very much for your comprehensive review and valuable feedback. We address your comments one by one as following:
>
> > **Q1: Your concerns can be summarized into the following two question. Here are our clarifications in details, respectively:**
>
> 1\) **Significance in novelty and improvements from existing work (PCGrad).**
>
> Firstly, although it's true that our work provides a projection-based approach that is originated and improved from PCGrad, as already stated in our paper, our work not only seeks to solve conflicts via the improved projection method, but also explores the advantages of achieving non-conflicting gradients and seeks to fully utilize such advantages for flexible and effective trade-offs among the tasks. Our main contributions are:
>
> - Introduction of the concept of strong non-confliction.
>
> - Demonstration that achieving strong non-confliction allows to effectively perform different trade-offs among the tasks via simple non-negative linear combination of the deconflicted gradients.
>
> Secondly, although PCGrad can be generalized to case of $T>2$, conflicts between the modified gradients $g_i'$ produced by PCGrad and the original $g_j$ become more and more frequent, indicating less effective performance. A simulation is designed to demonstrate the frequency of such conflicts when applying PCGrad to the case of $T>2$. Given task num $T$, each $g_i$ with dimension 100 is randomly generated IID from $U(-1,1)$ , and PCGrad is applied to generate $g_i'$. This experiment is repeated 10000 times for each $T$,  ratios of whether $g_1'$ conflicts with any of $\{g_2,...,g_T\}$ is recorded and presented below. Notice that with 6 tasks, about 58% of PCGrad modified gradients will still conflicts with other task gradients, which makes us, at least to some extent, less confident in performance of PCGrad.
>
> | Task Number        | 2     | 3      | 4      | 5      | 6      | 7      | 8      | 9      | 10     |
> |--------------------|-------|--------|--------|--------|--------|--------|--------|--------|--------|
> | Ratio of Conflicts | 0.00% | 13.23% | 29.59% | 44.88% | 58.38% | 69.45% | 76.71% | 82.60% | 86.68% |
> ||
>
>
> Besides, we also applied similar trade-off strategies on PCGrad for comparison. Note that best Average AUC of PCGrad (0.9384) is still less than best of GradOPS (0.9396), which is already significant enough for this dataset (averaged AUC gains achieved by [1] on UCI dataset over second best method barely make it to 0.001). Since PCGrad fails to guarantee strong non-conflicting gradients, such results are expected.
>
> |                    | Income | Marital | Education | Average |
> |--------------------|--------|---------|-----------|---------|
> | Single-task        | 0.9454 | 0.9817  | 0.8875    | 0.9382  |
> | PCGrad (alpha=0)   | 0.9428 | 0.9812  | 0.8856    | 0.9365  |
> | PCGrad (alpha=-1)  | 0.9451 | 0.9849  | 0.8852    | 0.9384  |
> | PCGrad (alpha=-2)  | 0.9449 | 0.9838  | 0.8846    | 0.9378  |
> | PCGrad (alpha=-3)  | 0.9446 | 0.9839  | 0.8830    | 0.9372  |
> | GradOPS (alpha=0)  | 0.9436 | 0.9817  | 0.8861    | 0.9371  |
> | GradOPS (alpha=-3) | 0.9459 | 0.9854  | 0.8876    | 0.9396  |
> ||
>
>
> Lastly, we would also like to re-emphasize that:
> - such simple and effective trade-offs among the tasks via non-negative linear combinations of task gradients are only available given strong non-conflicting gradients. As commented by other reviewers: "It is the flexible re-weighting that makes it interestingly competitive".
> - we also believe that ensuring strong non-conflicting gradients and studying more delicate and efficient trade-off strategies shall be an interesting and promising future research direction in MTL.
>
>
> 2\) **More theoretical analysis on GradOPS expected.**
>
> Firstly, some of the theoretical analyses in PCGrad are also applicable to GradOPS, at least when $T=2$. E.g. Theorem 2 on lower bound of the multi-task curvature and upper bound for loss, as well as the sufficient and necessary conditions for loss improvement in Theorem 3.
>
> Secondly, for GradOPS specifically, in future revisions we will also include in our Appendix a detailed analysis on effectiveness of GradOPS in achieving non-trivial strong non-conflicting gradients with supports on simulation and empirical experiments.
>
> &nbsp;
>
> [1] Modeling task relationships in multi-task learning with multi-gate mixture-of-experts, KDD 2018

---

### Official Review · Reviewer_JvTK · 2022-10-28

**Confidence:** 4
**Clarity, Quality, Novelty And Reproducibility:** Good
**Correctness:** 3
**Technical Novelty And Significance:** 3
**Empirical Novelty And Significance:** 3
**Recommendation:** 6

**Strength And Weaknesses:**

Strength
+ The proposed method produces deconflicted gradients that are not in confliction with any original gradients.
+ The deconflicted gradients make it much easy to define an overall updating gradient that does not conflict with any individual task specific gradient while weighting tasks during learning.
+ Code will be provided
+ Presentation of the manuscript is clear

Weakness
-  Regarding the discussion underneath Eq. (2), since g_i’ is located in the subspace that is orthogonal to that spanned by all g_j’s, should be g_i’• g_j = 0?
- The underlying assumption of existence of all g_i’ is that all g_i’s should be linearly independent among them. A discussion of such assumption in reality would strength the approach.
-  Resulting from using different \alpha values on test set are provided. However, no discussion is made how \alpha would be tuned in practice. Use results from the \alpha value leading to the best test performance to compare with other methods is not a fair comparison.
- Recent related work should be discussed and compared, for example:
Javaloy, A., & Valera, I. (2022). RotoGrad: Gradient Homogenization in Multitask Learning. ICLR, 1–24. http://arxiv.org/abs/2103.02631
- My biggest concern of this paper is the weak empirical results, with minimal difference in the results among compared the methods (especially, table 1 and 3), which does not support the advantage of the proposed method claimed by the authors.


**Summary Of The Paper:**

To address the task interference in multi-task learning, this paper presents an innovative method to obtain deconflicted gradients among tasks. This is done through projecting the gradient of a task to the subspace that is orthogonal to that spanned by the gradients of all other tasks. The authors provide the convergence analysis of the method and tested the method using two benchmark datasets and one large-scale recommendation dataset.

**Summary Of The Review:**

The idea is interesting and the method is technically sound. However, several weaknesses as listed, especially, the weak empirical results dampened my enthusiasm of this paper.

---

> ### Author Response · Authors · 2022-11-18
> **Author Response Part 2, responses to Q4 and Q5**
>
> > **Q4: discussion and comparison with recent works like RotoGrad.**
>
> Our paper is updated and recent methods like RotoGrad are referenced and discussed in Paragraph 2 of Section 5 Related Work. Here we would like to provide more details and discussions.
>
> 1\) We believe GradOPS is more generally applicable compared with RotoGrad for two reasons:
>
> - With strong non-conflicting gradient ensured, GradOPS is capable of searching different Pareto stationary points toward different trade-offs via a simple yet flexible re-weighting strategy. However, for RotoGrad, neither convergence to Pareto stationary points is guaranteed, nor the ability of allowing flexible trade-offs among the tasks is supported.
>
> - RotoGrad mainly focuses on situations where models have a shared backbone which produces a common latent representation across tasks. This indicates that they are mostly effective when the inputs for different tasks are shared and only the prediction output differs. In contrast, projection based methods like GradOPS and PCGrad, and MOO methods like MGDA and CAGrad are more generally applicable, reguardless of the model architecture or task inputs as long as there are shared structures.
>
> 2\) Results of RotoGrad (Rotate Only) on UCI dataset is added and presented below. GradOPS outperforms RotoGrad (Rotate Only) on all three tasks.
>
> |                       | Income | Marital | Education | Average |
> |-----------------------|--------|---------|-----------|---------|
> | Single-task           | 0.9454 | 0.9817  | 0.8875    | 0.9382  |
> | RotoGrad (Rotate Only)| 0.9430 | 0.9816  | 0.8854    | 0.9367  |
> | GradOPS (alpha=0)     | 0.9436 | 0.9817  | 0.8861    | 0.9371  |
> | GradOPS (alpha=-3)    | 0.9459 | 0.9854  | 0.8876    | 0.9396  |
> ||
>
> > **Q5: Significance of empirical results.**
>
> We would like to clarify that the improvements achieved by GradOPS are significant despite the fact that the absolute gains achieved on the metrics don't look great. As reference, averaged AUC gains achieved by GradOPS over second best method on UCI dataset is already greater than that achieved by [1]. We would also like to mention that gains on △m achieved by GradOPS on NYUv2 dataset in Table2 is also greater than that achieved by CAGrad.
>
> &nbsp;
>
> [1] Modeling task relationships in multi-task learning with multi-gate mixture-of-experts, KDD 2018

---

> > ### Comment · Reviewer_JvTK · 2022-12-12
> > **Re: Author resposne**
> >
> > Thanks to the authors for taking time to address my comments. As indicated in my initial review, the proposed method has merit in technical aspect. However, my main concern, the minimal difference in the performance comparing to existing method, remains. In addition, the authors studied the behavior of their method with varying alpha, but I am still not clear how alpha is chosen in practice.

---

> ### Author Response · Authors · 2022-11-18
> **Author Response Part 1, responses to Q1 - Q3**
>
> We thank the reviewer for the valuable feedback. We address the reviewer's concerns in the following.
>
> > **Q1: clarification on generation of $g_i'$.**
>
> As stated in line 4:8 of Algorithm 1, value of $g_i'$ depends on whether $g_i$ conflicts with each of $\\{g_j\\}_{j \neq i}$.
>
> If there is no conflicts between $g_i$ and each of $\\{g_j\\}_{j \neq i}$, we obtain $g_i'= g_i$ and thus $g_i' \cdot g_j \geq 0$.
>
> Otherwise, $g_i'$ is located in the subspace that is orthogonal to that spanned by all $\\{g_j\\}_{j \neq i}$, and thus $g_i' \cdot g_j = 0$ as you mentioned.
>
> > **Q2: Frequency of producing linearly correlated $\\{g_i'\\}$ in reality.**
>
> It's truth that the effectiveness of GradOPS can be limited when $\\{g_i'\\}$ are linearly correlated. However, we confirmed that such case is not observed in our simulation and real experiments:
>
> - In simulation, we randomly generate gradients with dimension 100 (much smaller than in real situation) for 10 tasks for 10000 times, all $\\{g_i\\}$ sampled are linearly independent, so is $\\{g_i'\\}$.
>
> - On both UCI and NYUv2 datasets, all $\\{g_i'\\}$ are also linearly independent with each other during training since there are only 3 tasks but gradients with large dimensions.
>
> We will also include a more rigorous analysis on linear relationships among $\\{g_i'\\}$ and $\\{g_i\\}$ into Appendix in future revision of the paper.
>
> > **Q3: Discussion on tuning $\alpha$ in practice and comparability of best performance with other methods.**
>
> Firstly, discussions on how $\alpha$ can make different trade-offs among the tasks are presented at the end of Section 3.2 and testified empirically in Appendix C. The major points are:
>
> - By checking if $r_i > 1$ or not, all tasks can be classified as dominating or dominated ones. Generally speaking, applying positive $\alpha$ will enforce the dominating tasks, and negative $\alpha$ will produce more balanced performance and enhance the dominated tasks (see Figure 6 in Appendix C)
>
> - In experiments, performance of using different $\alpha$ are presented to mainly demonstrate empirically the effectiveness of trade-offs among the tasks of GradOPS. Results and conclusions on how $\alpha$ yields different trade-offs are all consistent with our analysis in Section 3.2 as well as among the experiments in Section 4.
>
> - We would also like to emphasize that such simple and effective trade-offs among the tasks via non-negative linear combinations of task gradients are one of our major advantages, as commented by other reviewers that  "It is the flexible re-weighting that makes it interestingly competitive", and such ability is only available given strong non-conflicting gradients.
>
> - We also believe that ensuring strong non-conflicting gradients and studying more delicate and efficient trade-off strategies shall be an interesting and promising future research direction in MTL.
>
> Secondly, results of GradOPS and other methods are still comparable since:
>
> - Best hyperparameters are already applied for baseline methods which also need hyperparameters (GradNorm, CAGrad). On NYUv2 experiment, the best choice provided by the original paper is used, and on UCI dataset, parameters are searched according to strategies suggested by the papers and the one with best performance are selected. Details on how we determine values of these hyperparameters for GradNorm and CAGrad are added in Appendix B, and readers are notified at the beginning of Section 4 Experiments.
>
> - Note that GradOPS ($\alpha=0$) is always equally comparable with PCGrad.  Moreover, according to suggestions from other reviewers, we also testified that similar trade-off strategy is not as effective on PCGrad as on GradOPS (see table below). Best Average AUC of PCGrad (0.9384) is still less than best of GradOPS (0.9396), which is already significant enough on this dataset (averaged AUC gains achieved by [1] on UCI dataset over second best method barely make it to 0.001).
>
> &nbsp;
>
> [1] Modeling task relationships in multi-task learning with multi-gate mixture-of-experts, KDD 2018

---

### Official Review · Reviewer_wEFB · 2022-10-30

**Confidence:** 4
**Correctness:** 4
**Technical Novelty And Significance:** 2
**Empirical Novelty And Significance:** 3
**Recommendation:** 6

**Clarity, Quality, Novelty And Reproducibility:**

The paper is well-written and the idea is clearly illustrated with visualizations and descriptions of how the algorithm works. The experiments are quite extensive, the method is compared with several state-of-the art baselines. The idea of projecting gradient is not completely new, but more based on Yu et al. and extending it to projection to orthogonal subspace.

**Strength And Weaknesses:**

Strength:
1. The method proposed is guaranteed to resolve all gradient conflicts.
2. The method allows an easy way to control the weights of gradients from different tasks, which is nice to have to explore different Pareto optimal solutions to the problem.

Weaknesses:
1. The idea of projection is not new but based on Yu et al. 2022.
2. Not sure if strictly enforcing no gradient conflicts by projecting to the orthogonal subspace loses too much information about the gradient. In the worst-case scenario where there are a lot of gradient conflicts, this method will destroy a lot of the original gradient information by iterative projecting the gradients to a new orthogonal subspace. I would be more convinced if there is some evidence on how this procedure destroys or keeps most of the gradient information as compared to Yu et al.
3. I would love to see how Yu et al. performs with a similar kind of gradient re-weighting, though it would be non-trivial since there could be gradient conflicts in Yu et al.’s method (i.e. not all cosine similarity between g_i and G are positive, maybe use an exponential of the cosine similarity). As it seems like when alpha = 0, (i.e. gradient averaging after resolving the gradient conflicts), the proposed method does not have a clear advantage over Yu et al. It is the flexible re-weighting that makes it interestingly competitive. I wonder if that is also true by applying a similar re-weighting idea to Yu et al.

**Summary Of The Paper:**

This paper proposes a new method for resolving the gradient conflicts encountered in multi-task learning problems. Gradient conflict means when two gradients have a negative cosine similarity. The idea is based on [1] to project gradients of tasks to be orthogonal to all the other tasks if there is any gradient conflict between the two tasks. The difference is that in [1] gradient is projected iteratively based on pairs of tasks. In this work, the gradient is projected to the subspace that is orthogonal to the span of all other gradients. The paper also explores tuning the weight of gradient aggregation after resolving all gradient conflicts, so that it allows controlling the balance over different tasks. The experiments show that the proposed method is able to optimize for different Pareto optimal solutions and the results are competitive.

[1] Tianhe Yu, Saurabh Kumar, Abhishek Gupta, Sergey Levine, Karol Hausman, and Chelsea Finn. Gradient surgery for multi-task learning. Advances in Neural Information Processing Systems, 33:5824–5836, 2020.

**Summary Of The Review:**

Overall, I think the paper presents an interesting perspective on how to do flexible re-weighting of gradients based on the cosine similarity of the independent task gradient and the aggregated gradient. The idea of gradient projection is not completely new but inherited from Yu et al. I have some reservations about whether it makes the most sense to project gradient to be orthogonal to all other gradients to resolve the conflict, as it can lose a lot of the gradient information.

---

> ### Author Response · Authors · 2022-11-18
> **Author Response Part 2, responses to Q3**
>
> > **Q3: comparison of trade-off effectiveness with PCGrad applying similar re-weighting strategy.**
>
> We appreciate your suggestion that replacing $R_i$ in Eq.(4) by $\exp(R_i)$ for PCGrad since $R_i$ can be negative in PCGrad, and such replacement will also grant similar trade-off abilities to PCGrad. On similar re-weighting idea applied to PCGrad (Yu et al.), results are provided in the following table.
>
> - Overall performance: although PCGrad with negative $\alpha$  (-1, -2, -3) also achieved improvement on Income, Marital and Average AUCs (compared to PCGrad ($\alpha=0$)), the best Average AUC of PCGrad (0.9384) is still less than the best of GradOPS (0.9396), which is already significant enough for this dataset (averaged AUC gains achieved by [1] on UCI dataset over second best method barely make it to 0.001). Since PCGrad fails to guarantee strong non-conflicting gradients, such results are expected.
>
> - Trade-off effectiveness: we really agree with your understanding that "It is the flexible re-weighting that makes it interestingly competitive", and such advantage is achieved via strong non-conflicition guarantees. Re-weighting strategy applied on GradOPS is more efficient than similar strategy on PCGrad. Specifically, PCGrad ($\alpha=-1$) achieved best Average AUC at the expense of Task Education, whereas similar trade-off strategy works much better on GradOPS ($\alpha=-3$) and even no task are sacrificed, since GradOPS ensures that the re-weightings are performed base on strong non-conflicting gradients.
>
> This experiment will also be added to our Appendix in the revision of the paper.
>
>
>
> |                    | Income | Marital | Education | Average |
> |--------------------|--------|---------|-----------|---------|
> | Single-task        | 0.9454 | 0.9817  | 0.8875    | 0.9382  |
> | PCGrad (alpha=0)   | 0.9428 | 0.9812  | 0.8856    | 0.9365  |
> | PCGrad (alpha=-1)  | 0.9451 | 0.9849  | 0.8852    | 0.9384  |
> | PCGrad (alpha=-2)  | 0.9449 | 0.9838  | 0.8846    | 0.9378  |
> | PCGrad (alpha=-3)  | 0.9446 | 0.9839  | 0.8830    | 0.9372  |
> | GradOPS (alpha=0)  | 0.9436 | 0.9817  | 0.8861    | 0.9371  |
> | GradOPS (alpha=-3) | 0.9459 | 0.9854  | 0.8876    | 0.9396  |
> ||
>
>
> &nbsp;
>
> [1] Modeling task relationships in multi-task learning with multi-gate mixture-of-experts, KDD 2018

---

> ### Author Response · Authors · 2022-11-18
> **Author Response Part 1, responses to Q1 and Q2**
>
> Thank you for your comments and suggestions. We would like to response to your concerns in detail.
>
> > **Q1: The idea of projection is not new but based on Yu et al.**
>
> It's true that the idea of projection is based on PCGrad (Yu et al.), as already stated in our paper, and our work not only seeks to solve conflicts via improved projection method, but also utilizes the advantages of achieving non-conflicting gradients. Our main contributions are:
>
> - Introduction of the concept of strong non-confliction,
> - Demonstration that strong non-confliction allows to effectively perform different trade-offs among the tasks via simple non-negative linear combination of the deconflicted gradients.
>
> Besides, as we will stated later in answering Q3, without strong non-confliction guarantees, similar gradient re-weighting strategy applied on PCGrad is less effective. We also perform extensive analysis and experiments on how different $\alpha$ in our GradOPS can effectively make different trade-offs among the dominating and dominated tasks (see Discussion about $\alpha$ in Section 3.2 and Appendix C). In all, we believe that ensuring strong non-conflicting gradients and studying more delicate and efficient trade-off strategies shall be an interesting and promising future research direction in MTL.
>
> > **Q2: concerns on whether GradOPS loses too much gradient information.**
>
> Your noticing is appreciated. At the beginning of our research, we also concern about whether GradOPS drops too much gradient information, and till now it's still one of our lasting research topic how to keep the most non-conflicting gradient information. Thankfully, our extensive experiments on different domains demonstrate that, current version of GradOPS already keeps sufficient information and is already capable of performing flexible trade-offs among the tasks as demonstrated. If performance on validation datasets can be regarded as a measure of positive information extraction, GradOPS is quite efficient and GradOPS ($\alpha=0$) always works better than PCGrad. We would also like to state that:
>
> - When defining the loss of gradient information, we shouldn't just focus on comparing each $g'_i$ with $g_i$ but pay more attention to comparing of $G$ or $G'$ with each $g_i$, since $G$ or $G'$ is the final aggregated update direction. It's hard to compare whether GradOPS drops more information than PCGrad / the vanilla method or not, since both PCGrad and the vanilla method are expected to loss significant information during the aggregation operation without strong non-confliction guarantees.
>
> - We also find it hard to design fair metrics evaluating the loss of information because:
>
>     - If we choose to compare the similarity between each modified $g'_i$ and original $g_i$ as the ability to preserve gradient information, then 1) the vanilla method keeps the most information but shows poor performance, and 2) not all information of original $g_i$ are useful and non-conflicting with each other.
>
>     - From our view, only non-conflicting information is useful, which is already our goal to keep the most non-conflicting gradient subparts. It's one of our future works to find and prove the best algorithm that keep the most non-conflicting gradient information.
>
>     - If you have any suggestion on fair metrics to better evaluate the loss of gradient information, please notify us and we would add related analysis to the Appendix.

---

### Author Response · Authors · 2022-11-17
**Updated manuscript**

Dear reviewers and AC,

We would like to inform you that we have updated the manuscript, trying to incorporate as much feedback as possible. The most important changes are the following:
- We have updated Section 5 Related Work to include discussions on references stated by the reviewers.
- A more detailed description about hyperparameters for GradNorm and CAGrad are provided in Appendix B.
- To be more rigorous, assumptions of the convergence of the proposed algorithm in our theorems are expressed explicitly in our paper.
- As suggested, for better clarity and beauty, some notations in Figure 1 are slightly modified, and Eq.(1) is slightly modified to avoid ambiguity.
- An expanded version of Table 1 is provided in Supplementary Material, in which results of experiments on UCI dataset are appended with standard deviations as well as significance of improvements in hypothesis testings.
- A primal version of GradOPS source code is provided in Supplementary Material. Formal version of both GradOPS and experiments will be released later.

We are working on further improving our manuscript. We will respond to individual reviewers to address their specific concerns separately. Thanks.

---

### Author Response · Authors · 2022-12-02
**To all reviewers**

We sincerely thank all the reviewers for your comments and suggestions, we have already provided details response to all the questions and uploaded our revised manuscript.

Since the discussion period is coming to an end, we would like to know if you have any further questions. From our point of view, we feel that we have clarified the concerns and misunderstandings that you have raised in the initial review.
If you could re-assess our paper based on our response, let us know if you might find our work more positive, we would be very grateful!

If you have any further questions, please ask without hesitation and we are always available for further clarification.

Best,

---

### Public Comment · ~Zhi_Song1 · 2023-02-03
**Severe Problems with This Paper**

The paper has an error in the theoretical result. I highly recommend the author fix this problem in the future version.

At the end of Page 13. The authors claim that $g_i = 0$ or $g_i$ belongs to a subspace $\text{span}(g_j),j\not = i$. This can not lead to the result in the sentence: "This means that there exists a convex combination of the gradients $\{g_i\}$ at this point $\theta^*$ that equals zero". In fact, if $g_i$ belongs to that subspace, we can only say there exists a non-zero linear combination that $\sum_{i=1}^m \alpha_i g_i = 0$ ($\alpha_i\not=0$). The author can verify this in Wikipedia https://en.wikipedia.org/wiki/Linear_subspace.

A simple example may help understand the error of this proof. Suppose $g_1=(-1,0)$, $g_2=(1,1)$ and $g_3=(0,1)$. Clearly, there exist no $\alpha \in \Delta^2$ that $\sum_{i=1}^3\alpha_i g_i=0$, where $\Delta^2$ represents a two dimensional simplex. So, in this case, we do not reach the Pareto stationary point.  However, according to the method proposed in this paper, $g_1$ and $g_2$ are in conflict, so we need to do the gradient projection for $g_1$. We have $u_1= (-1/2,1/2)$ and $u_2= (1,1)$ and we further obtain $g'_1=0$. Thus we reach the Pareto stationary point according to the proof on Page 13. This contradicts the facts.
Thus the convergence proof provided in the Appendix seems not correct. At least. $g'_i$ converge to zero can not reflect anything.

---

> ### Author Response · Authors · 2023-02-03
> **Author Response**
>
> Thanks for pointing out the error of theoretical analysis in Theorem1.
>
> Firstly, we already identified and fixed the error during our continual refinement of our paper. For now, in latest version of our paper, this problem can be avoided simply by applying MMO methods like MGDA[1] at the expense of losing flexible trade-offs among the tasks under such situation $g'_i = \textbf{0}, \forall i$. We are very willing to provide the latest version for further discussing.
>
> Currently, we are still working on looking for better gradient deconfliction methods that support flexible trade-offs under such situation with convergence guarantees.
>
> Lastly, we would also like to restate that such extreme case happens only when $g_i' = \textbf{0}$ for all GradOPS-deconflicted gradients, which rarely happens and never observed in all experiments in Section 4, thus experiment results and major conclusions are not affected.
>
> [1] Multiple-gradient descent algorithm (MGDA) for multiobjective optimization, Comptes Rendus Mathematique 2012

---

### Decision · Program_Chairs · 2023-01-20

**Decision:**

Reject

**Justification For Why Not Higher Score:**

There are some concerns about the technical design of the proposed method. Experiments are not convincing.

**Justification For Why Not Lower Score:**

N/A

**Metareview: Summary, Strengths And Weaknesses:**

In this paper, the authors proposed a new multi-task learning method to avoid conflicts of gradients among tasks and search for diverse solutions towards different trade-off preferences among them. Though the proposed idea is potentially interesting, there are some major concerns about some technical details of the proposed method. These concerns remain after rebuttal. Moreover, the experimental results are not convincing to demonstrate the superior performance of the proposed method over existing methods.

In summary, this work is potentially promising work, but is not ready to be published based on its current shape.